# Insulin action and resistance are dependent on a GSK3β-FBXW7-ERRα transcriptional axis

Hui Xia[1,2], Charlotte Scholtes[1], Catherine R. Dufour[1], Carlo Ouellet[1], Majid Ghahremani[1] & Vincent Giguère [1,2✉]

Insulin resistance, a harbinger of the metabolic syndrome, is a state of compromised hormonal response resulting from the dysregulation of a wide range of insulin-controlled cellular processes. However, how insulin affects cellular energy metabolism via long-term transcriptional regulation and whether boosting mitochondrial function alleviates insulin resistance remains to be elucidated. Herein we reveal that insulin directly enhances the activity of the nuclear receptor ERRα via a GSK3β/FBXW7 signaling axis. Liver-specific deletion of GSK3β or FBXW7 and mice harboring mutations of ERRα phosphosites (ERRα[3SA]) co-targeted by GSK3β/FBXW7 result in accumulated ERRα proteins that no longer respond to fluctuating insulin levels. ERRα[3SA] mice display reprogrammed liver and muscle transcriptomes, resulting in compromised energy homeostasis and reduced insulin sensitivity despite improved mitochondrial function. This crossroad of insulin signaling and transcriptional control by a nuclear receptor offers a framework to better understand the complex cellular processes contributing to the development of insulin resistance.

[1] Rosalind and Morris Goodman Cancer Research Institute, McGill University, Montréal, QC H3A 1A3, Canada. [2] Department of Biochemistry, Faculty of Medicine and Health Sciences, McGill University, Montréal, QC H3G 1Y6, Canada. ✉email: vincent.giguere@mcgill.ca

Humans have evolved to metabolically adapt to unpredictable diets by storing calories as fat when well-fed and burning fat during food shortage to maintain euglycemia. This intricate control upon fasting and feeding is predominantly governed by insulin, which is the primary glucose-lowering and fat-conserving hormone[1]. Insulin is uniquely required for anabolic processes like lipogenesis, glycogenesis and protein synthesis in adipose, liver and muscle, and for simultaneous suppression of adipose lipolysis, hepatic glucose production, and protein breakdown[2]. Insulin resistance instills when target tissues lose their responsiveness to the hormone and has long been considered to be central to the pathophysiology of metabolic disorders such as type 2 diabetes (T2D), obesity, cardiovascular disease, and non-alcoholic fatty liver disease (NAFLD)[3].

Insulin resistance develops from defects in multiple tissues and the intracellular mechanisms are highly complex. While the molecular events underlying insulin action have been substantially studied, major gaps still persist in our full comprehension of the downstream mechanisms involved in insulin signaling[2]. Given that many signaling pathways eventually converge at the transcriptional level, gene dysregulation is among one of the main causes of insulin resistance[4,5]. Transcriptional regulation is crucial in the fasting-feeding transition by turning on and off thousands of genes involved in fuel metabolism. Indeed, transcription factors such as FoxO1 and SREBP1 have been identified as regulators of insulin action and participants in the progression of insulin resistance via their control of the expression of specific genes in glucogenic and lipogenic pathways[2,4,5].

ERRα is an orphan nuclear receptor best known for controlling energy homeostasis[6,7] and playing a major role in mitochondrial biogenesis and function[8–11]. Global ERRα-null mice are lean and are protected from diet-induced obesity (DIO) and insulin resistance[12,13], and ERRα-targeted oxidative phosphorylation (OXPHOS) genes are downregulated in human diabetic muscle, suggesting the potential of modulating ERRα activity to treat individuals with T2D[14,15]. Nonetheless, whether ERRα is a direct modulator of insulin action is currently unknown. Here, we uncover an insulin-dependent sequential post-translational regulatory event modulating ERRα stability and activity, in which ERRα phosphorylation by glycogen synthase kinase 3 beta (GSK3β) promotes its recognition and ubiquitination by the E3 ligase, FBXW7, resulting in ERRα nuclear export and subsequent degradation via the proteasome. Specific interruption of the GSK3β/FBXW7/ERRα cascade by genetically silencing the GSK3β-targeted ERRα phosphosites in vivo results in the persistent stabilization of ERRα protein thus altering insulin-responsive transcriptomes, eventually leading to insulin resistance, notwithstanding improved mitochondrial function. Our findings thus uncover and highlight the importance of an insulin-dependent transcriptional pathway that can be sequentially targeted and reveal ERRα as a central partner in insulin action and resistance.

## Results

**ERRα is an insulin-stimulated transcription factor.** To investigate whether ERRα acts as a nutrient sensor, we examined the expression of ERRα under different nutritional statuses. We observed elevated hepatic ERRα protein levels without affecting its mRNA expression in response to increased nutrients intake upon fasting/refeeding transition, acute high fat-feeding and during the dark phase of the circadian cycle when mice consume food, all behaviors accompanied by activation of the insulin signaling cascade (Fig. 1a–f). We further demonstrated that ERRα protein but not mRNA levels were similarly induced by direct administration of glucose and insulin in vivo (Fig. 1g–j). Consistently, stimulation of HepG2 cells with either glucose or insulin

increased ERRα protein levels in a dose-dependent manner without altering mRNA expression (Supplementary Fig. 1a–d). A time-course experiment in hepatocytes showed that insulin rapidly stabilized ERRα protein but not mRNA in the nucleus as early as 15 min post-stimulation with the effect lasting at least 4 h (Supplementary Fig. 1e, f). Further, insulin increased the level of ERRα protein by approximately threefold in the nuclear fractionation of mouse liver without altering its mRNA (Fig. 1k, l).

To gain further insights into the impact of insulin-stimulated ERRα levels on gene expression, we next performed hepatic RNA sequencing in wild-type (WT) and ERRα$^{-/-}$ mice treated with or without insulin. Although a similar number of differentially expressed genes (DEGs) were induced by insulin in livers of WT and ERRα$^{-/-}$ mice, there was only a modest overlap, indicating that loss of ERRα reprograms the insulin-responsive transcriptome (Supplementary Fig. 1g). Notably, in WT mice, insulin stimulation resulted in 2,455 hepatic DEGs, of which ~67% (1,653) could not be modulated by insulin in ERRα$^{-/-}$ mice (Fig. 1m and Supplementary Data 1), implying that their regulation by insulin requires the presence of ERRα. For example, *Cpt1a* and *Pck1*, genes encoding rate-limiting enzymes in fatty acid oxidation and gluconeogenesis, respectively, were transcriptionally downregulated by insulin while genes encoding proteins involved in lipogenesis (*Cs*) and translation (*Eif1a*) were upregulated, all in an ERRα-dependent manner (Supplementary Fig. 1h and Supplementary Data 1). Further interrogation of these datasets revealed that insulin-regulated genes found differently expressed in WT and ERRα$^{-/-}$ mice upon insulin were mainly enriched in mitochondrial-related processes as well as glucose, lipid and amino acid metabolism (Fig. 1n, Supplementary Fig. 1i, and Supplementary Data 2). In particular, we found that a set of mitochondrial associated genes as well as proteins of OXPHOS subunits were upregulated by insulin in WT but not in ERRα$^{-/-}$ mice (Fig. 1n, Supplementary Fig. 1j). This result signifies the importance of ERRα in insulin-mediated mitochondrial biogenesis and OXPHOS, which is likely used to generate ATP, essential for biosynthetic processes. Consistently, intersection of transcriptome and ERRα liver ChIP-seq datasets[16] revealed that over 40% of the insulin-responsive gene signature in cultured hepatocytes is directly targeted by ERRα, which were also highly enriched in lipids and carbohydrate metabolism as well as in mitochondrial functions (Supplementary Fig. 1k and Supplementary Data 3). Of note, ERRα targets also showed great overlap with those bound by FoxO1 or SREBP1 in liver (Supplementary Fig. 1l and Supplementary Data 4), two well-known transcription factors downstream of the hepatic insulin signaling pathway[17]. Overall, this data demonstrates ERRα as a direct and potent insulin-responsive transcription factor.

**GSK3β is required for insulin-dependent ERRα stabilization.** We next sought to characterize how insulin stabilizes ERRα. The acute induction of ERRα protein by insulin without affecting its mRNA expression suggests the involvement of a post-transcriptional process. ERRα has been reported as a phosphoprotein that can be phosphorylated by specific protein kinases[6,18,19]. Using the predicting programs Scansite 4.0 (https://scansite4.mit.edu), PhosphoNet (http://www.phosphonet.ca) and GPS 3.0[20], ERRα was predicted to be phosphorylated by cyclin-dependent kinases (CDKs), glycogen synthase kinase (GSK)3, extracellular signal-regulated kinases (ERKs), casein kinases (CKs), and p38 mitogen-activated protein kinases (MAPKs). Hepatocytes treated with inhibitors targeting selected kinases showed that ERRα protein was best stabilized by the GSK3 inhibitor, followed by CK2 inhibition, with inhibitors against CDK2/5, p38 MAPK and CDK1 having no impact (Fig. 2a).

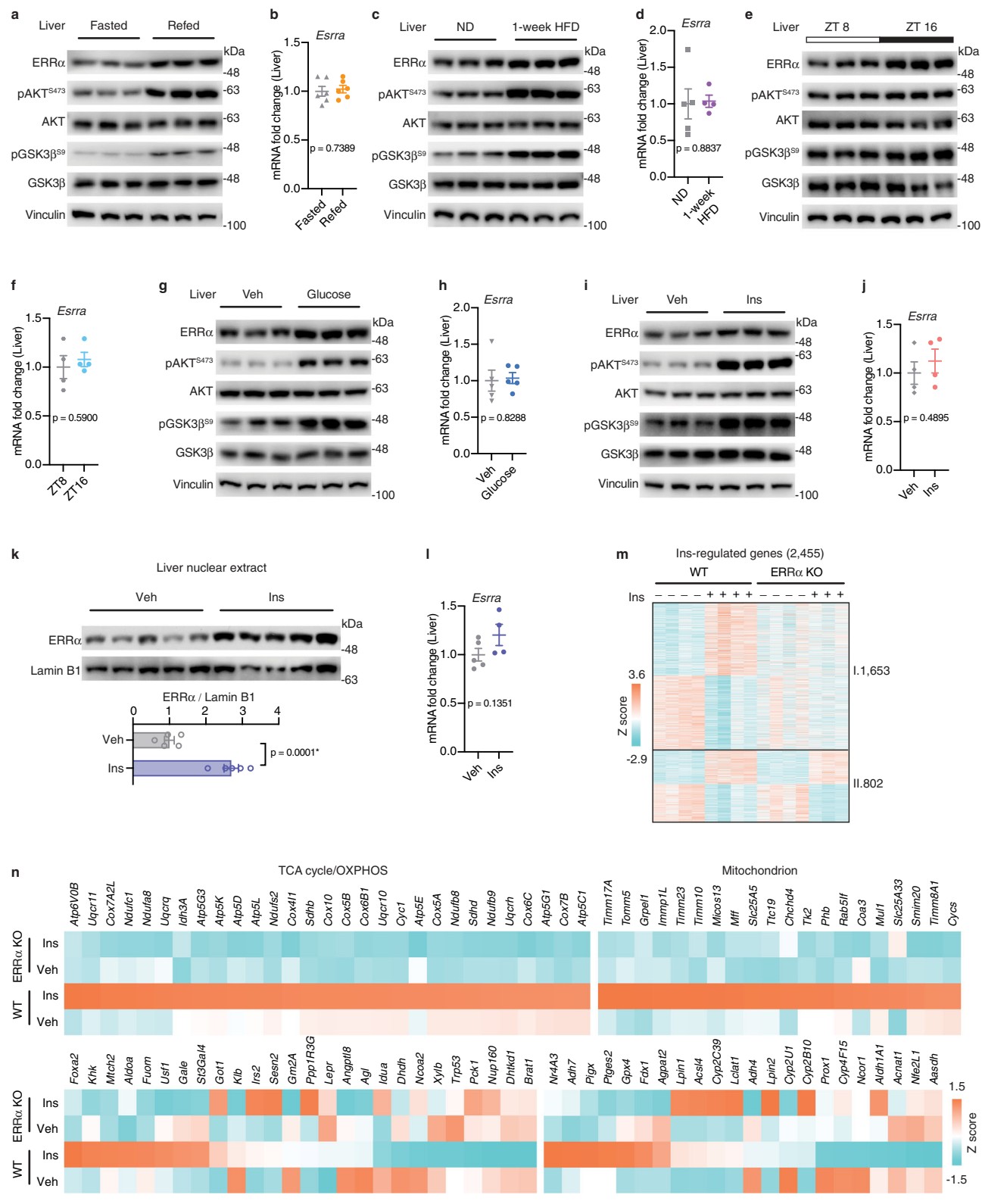

Introduction of siRNAs specifically targeting GSK3α, GSK3β and CK2 further demonstrated that the effect of insulin on ERRα stability was indeed replicated by inhibition of GSK3β (Fig. 2b, Supplementary Fig. 2a). LiCl, a classic GSK3β inhibitor, also stabilized ERRα protein (Supplementary Fig. 2b). Conversely, pharmacological inhibition of the PI3K/AKT/mTOR pathway by PI3K inhibitors (Wortmannin, LY294002), AKT inhibitors

(MK-2206, GDC-0068) and mTOR inhibitors (Rapamycin and Torin 1) all led to ERRα degradation via dephosphorylation and activation of GSK3β (Supplementary Fig. 2c). Consistently, genetic activation of GSK3β by overexpressing its constitutively active S9A mutant[21] also promoted ERRα degradation (Fig. 2c). To further investigate the impact of GSK3β loss on ERRα stability in vivo, we used the Cre-lox P system to specifically delete GSK3β

**Fig. 1 ERRα is an insulin-stimulated transcription factor. a–j** Hepatic ERRα protein (**a, c, e, g, i**) and mRNA (**b, d, f, h, j**) levels in corresponding nutritional states: **a, b** refeeding post overnight fasting; **c, d** normal diet (ND) or 1-week high-fat diet (HFD) feeding; **e, f** Mice sacrificed at ZT8 or ZT16; **g, h** Mice treated with vehicle (Veh) or glucose for 0.5 h after overnight fasting; **i, j** Mice treated with Veh or insulin (Ins) for 0.5 h after overnight fasting. **a, c, e, g, i** Each lane represents liver from one mouse, $n = 3$. **b** $n = 6$ per group. **d** $n = 5$ for ND, $n = 4$ for 1-week HFD. **f** $n = 4$ per group; **h** $n = 5$ per group; **j** $n = 4$ per group. **k** Immunoblots (up, each lane represents liver from one mouse, $n = 5$ per group) and quantification (down) of nuclear ERRα protein levels in livers from mice treated with insulin for 4 h post overnight fasting. **l** ERRα mRNA levels in livers from mice treated with insulin for 4 h post overnight fasting (Veh, $n = 5$; Ins, $n = 4$). **m** Z score heatmap of insulin-regulated hepatic genes identified from WT and ERRα$^{-/-}$ mice stimulated with insulin for 4 h post overnight fasting ($p < 0.05$, $|FC| \geq 1.20$). I. ERRα dependent & Ins-regulated DEGs: Genes significantly modulated by insulin in WT mice only. II. Other Ins-regulated DEGs: Genes modulated by insulin in both WT and ERRα$^{-/-}$ mice ($n = 3$ for ERRα KO Ins, $n = 4$ for the others). See also Supplementary Fig. 1h. **n** Z score heatmap representation of subsets of insulin-regulated hepatic genes differently expressed between WT and ERRα$^{-/-}$ mice upon insulin ($p < 0.05$; $|FC| \geq 1.20$; $n = 3$ for ERRα KO Ins, $n = 4$ for the others). Data are presented as means ± SEM, *$p < 0.05$, unpaired two-tailed Student's $t$ test (**b, d, f, h, j, k, l**). Source data are provided as a Source Data file.

in liver by breeding GSK3β floxed mice with Alb-Cre mice. Indeed, ERRα protein accumulated in livers of GSK3β liver-specific knockout (LKO) mice in comparison with their floxed littermates (Fig. 2d) or age-matched Alb-Cre mice (Supplementary Fig. 2d). These results clearly show that GSK3β negatively regulates ERRα protein levels in vitro and in vivo. Most importantly, in the absence of GSK3β, ERRα protein but not mRNA was accumulated and could not be further elevated by insulin or refeeding (Fig. 2e–h). Alternatively, re-expression of the constitutively active GSK3β S9A but not the kinase defective K85A mutant degraded the accumulated ERRα protein in the GSK3β knock-down cells (Fig. 2i, compare lanes 9, 11 to lanes 1, 7), and insulin had no further effects on ERRα protein stability once GSK3β could not be phosphorylated by insulin or lost its kinase activity (Fig. 2i, compare lanes 9, 10, 11, 12 to lanes 7, 8). Consistently, blocking the insulin signaling transduction to GSK3β by AKT inhibitors abrogated insulin-mediated accumulation of ERRα, in line with the release of GSK3β from insulin-mediated phosphorylation and suppression (Supplementary Fig. 2e). Together, our results demonstrate that GSK3β kinase activity is required for insulin control of ERRα protein stability through the PI3K/AKT cascade (Fig. 2j).

**ERRα phosphorylation by GSK3β leads to its proteasome-mediated degradation.** We previously showed that ERRα is phosphorylated at S19 and S22[18]. Of interest, we found significant changes of ERRα phosphorylation at these sites post insulin delivery in a recent high-throughput insulin-based mouse liver phosphoproteomic dataset[22] (Fig. 3a). We also noted that insulin-stimulated phosphorylation changes of GSK3β (inactive) followed that of its upstream kinase AKT (active), which reversely correlated with ERRα phospho-status at many time points (Fig. 3a). Indeed, impeding GSK3β activity indirectly by insulin stimulation or directly by genetic interference attenuates ERRα phosphorylation while stabilizing ERRα protein (Fig. 3b, c). In comparison with whole-cell extract (WCE), phos-tag gel examination of the nuclear extract from HepG2 cells could not reveal an upper phosphorylated ERRα band shift (Supplementary Fig. 3a), indicating that insulin-stimulated nuclear ERRα protein was dephosphorylated. To substantiate this observation, mutations were introduced to the insulin-sensitive S19 and S22 residues. Nuclear export of ERRα exclusively with the phospho-mimicking ERRα S19D + S22D mutant, while the phospho-deficient S19A + S22A mutant remained in the nucleus (Supplementary Fig. 3b), implying that ERRα translocation between the cytoplasm and nucleus is mediated by phosphorylation.

GSK3β phosphorylates proteins at serine or threonine that are usually located four residues apart (S/T-X-X-X-S/T–P), whereas the C-terminal S/T–P is a priming, pre-phosphorylated residue[23]. Bioinformatic analysis based on conserved kinase substrate motifs predicted the ERRα amino terminus to contain three GSK3β

consensus phosphorylation motifs consisting of phosphorylation sites S19, S22, S26, and C-terminal priming site E30, which are conserved across various species (Fig. 3d). The insulin-responsive ERRα S19 and S22 were within the GSK3β substrate motifs. Notably, ERRα phosphorylation at S19, S22, S26 are the most common ERRα post-translational modifications (PTMs) revealed by high-throughput mass spectrometry-based studies according to PhosphoSitePlus (http://www.phosphosite.org) (Supplementary Fig. 3c), indicating the critical importance of these modifications.

We next mutated the three encompassed serine residues individually or in combination. Six different in vitro-translated GST-ERRα proteins were tested in a kinase assay, validating that ERRα can be directly phosphorylated by GSK3β (Fig. 3e). Mutation of any of these three serine residues to alanine markedly decreased GSK3β-mediated ERRα phosphorylation (Fig. 3e). Alternatively, ERRα S26E mutant, by mimicking the C-terminal priming phosphorylation for S22, was phosphorylated by GSK3β to a similar extent to that of WT (Fig. 3e). Remarkably, the GSK3β-defective ERRα 3SA mutant (S19A + S22A + S26A) was resistant to both GSK3β-mediated degradation (Fig. 3f) and insulin-induced stabilization (Fig. 3g).

We then defined how ERRα protein stability is controlled by the insulin/GSK3β axis. Given the ubiquitin-proteasome system (UPS) is a major mechanism regulating protein turnover[24], we assessed and observed that MG132-mediated inhibition of the proteasome led to accumulation of ERRα protein without altering its mRNA level (Supplementary Fig. 3d, e). In the refed liver, ERRα ubiquitination level was markedly downregulated as opposed to its increased protein level (Fig. 3h). We further confirmed this finding in starved and insulin-stimulated hepatocytes (Supplementary Fig. 3f), reinforcing that insulin safeguards ERRα from ubiquitination and degradation. Likewise, GSK3β knockdown greatly decreased ERRα ubiquitination level (Fig. 3i), implicating that GSK3β facilitates ERRα ubiquitination and degradation. Furthermore, GSK3β-mediated ERRα degradation was rescued by inhibiting the proteasome (Fig. 3j). Together, these results show that insulin protects ERRα from GSK3β-mediated phosphorylation, thus attenuating ERRα ubiquitination and degradation (Fig. 3k).

**GSK3β-mediated ERRα degradation is dependent on FBXW7.** To identify the E3 ligase involved in insulin/GSK3β-mediated ERRα degradation, we devised a guided screening strategy as outlined in Supplementary Fig. 4a (See also Supplementary Data 5). Given that nuclear receptors usually control their own activities through feedback loops and the crucial role of ERRα in metabolism, we first identified 477 E3 ligases which were either direct ERRα targets or were involved in metabolic processes. Among those, Stub1 and Parkin, two previously reported ERRα E3 ligases were identified in that subgroup[16,25], denoting the reliability of the screen. We then selected 26 E3 ligases based on their tissue-specific

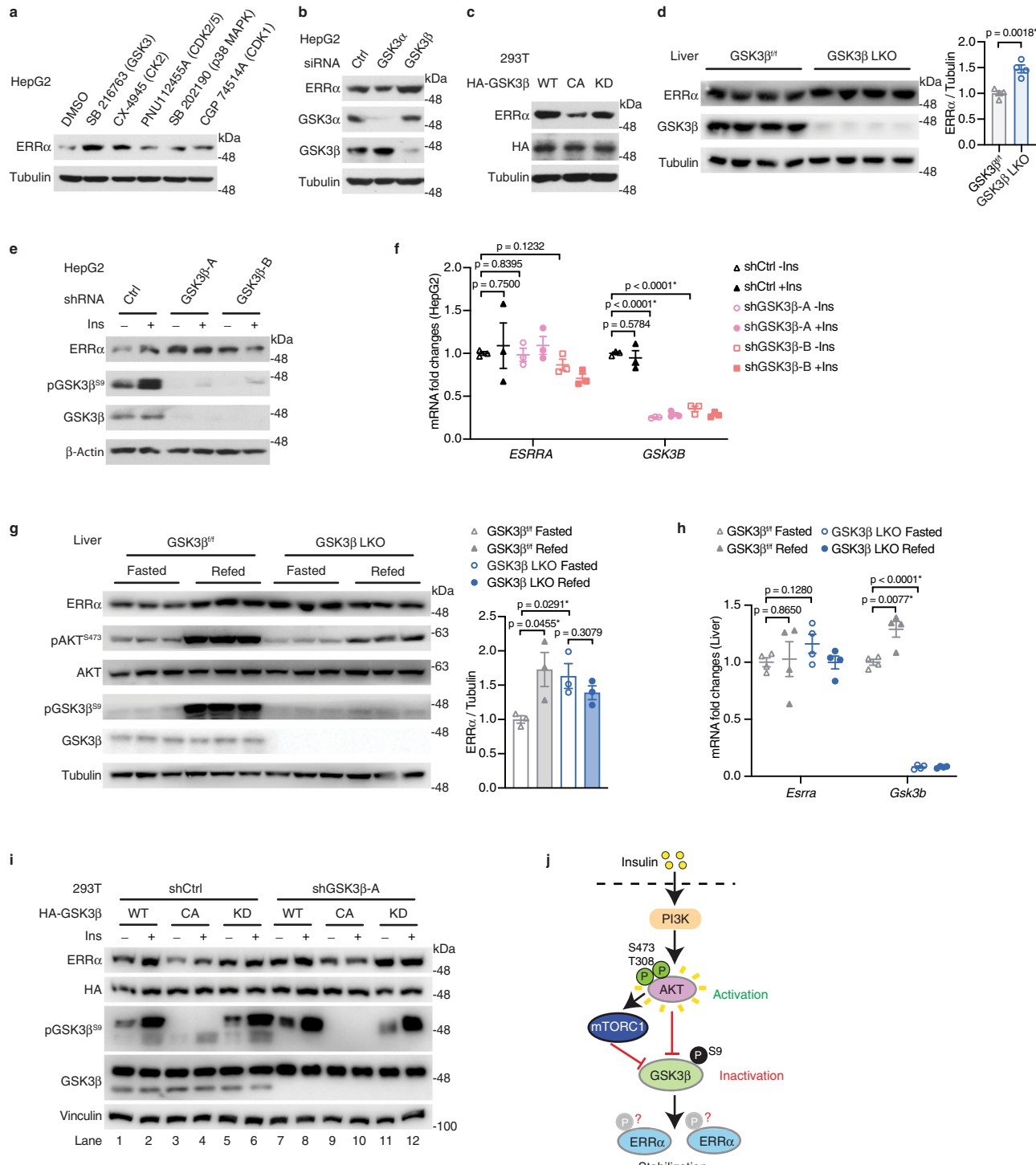

**Fig. 2 GSK3β is required for the control of ERRα stability by insulin. a** ERRα protein levels in HepG2 cells treated with either DMSO control or 10 μM of kinase specific-inhibitors SB 216763, CX-4945, PNU112455A, SB 202190 or CGP 74514A for 4 h. **b** ERRα protein levels in HepG2 cells transfected with 20 nM of either control, GSK3α or GSK3β siRNA for 72 h. **c** ERRα protein levels in 293T cells transfected with vectors expressing HA-GSK3β WT, constitutively active S9A mutant (CA), or kinase defective K85A mutant (KD) for 48 h. **d** Immunoblots (left, each lane represents liver from one mouse, $n = 4$ per group) and quantification (right) of ERRα protein levels in livers from GSK3β$^{f/f}$ and GSK3β LKO mice. **e** ERRα protein levels and **f** mRNA levels of *ESRRA* and *GSK3B* ($n = 3$ per group) in HepG2 cells stably expressing either control shRNA or 2 distinct shRNAs targeting GSK3β. Cells were treated with 10 μg/ml insulin for 30 min after overnight serum starvation. **g** Western blots (left, each lane represents liver from one mouse, $n = 3$ per group) and quantification (right) of ERRα protein levels in livers from GSK3β LKO and their floxed littermates upon a fasting-refeeding transition. **h** *Esrra* and *Gsk3b* mRNA levels in control and GSK3β-null livers upon a fasting-refeeding transition ($n = 4$ per group). **i** 293T cells stably expressing either control or shRNA targeting GSK3β were transfected with HA-GSK3β WT, CA, or KD vectors for 48 h, cells were treated with 10 μg/ml insulin for 30 min after overnight serum starvation. **j** Proposed model of insulin-mediated stabilization of ERRα protein through the PI3K/AKT/ GSK3β pathway. Data are presented as means ± SEM, *$p < 0.05$, unpaired two-tailed Student's $t$ test (**d**, **f–h**). Source data are provided as a Source Data file.

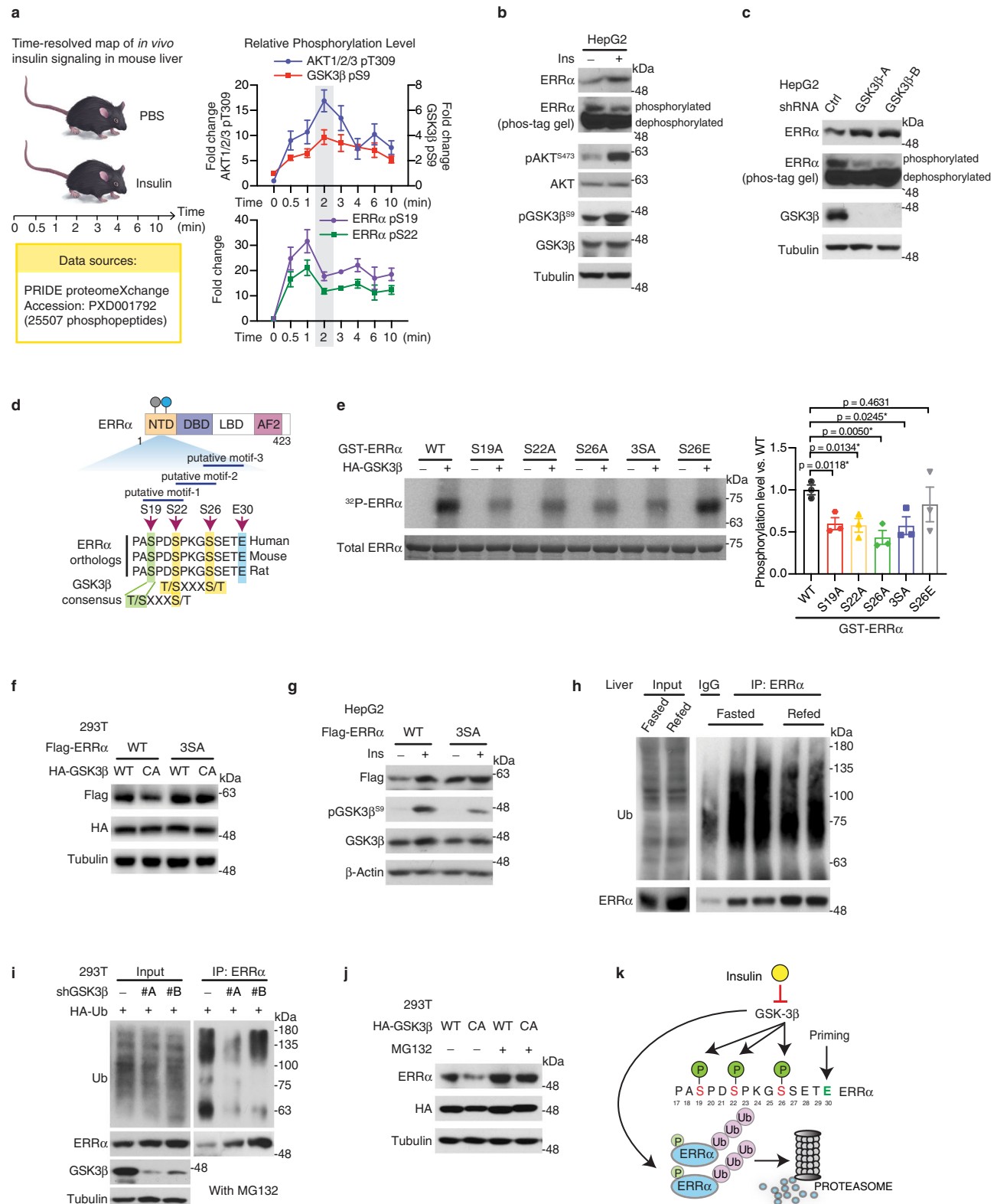

expression patterns, cellular localizations as well as known functions. These 26 E3s were then subjected to a second round of screening (Fig. 4a). Overexpression studies in HepG2 cells showed that 6 of the 26 E3s promoted ERRα degradation (Supplementary Fig. 4b) in an UPS-dependent manner (Supplementary Fig. 4c). These 6 E3s were also shown to decrease ERRα half-life (Supplementary Fig. 4d). Co-IP experiments confirmed the interaction between the 6 E3 candidates and ERRα (Supplementary Fig. 4e). Of these, 4 E3s were further demonstrated to enhance ERRα ubiquitination (Supplementary Fig. 4f), while their shRNA-mediated knockdown fostered ERRα accumulation (Supplementary Fig. 4g). Overall, the screen uncovered 4 E3 ligases (FBXO7, FBXO11, FBXW7α and WWP1) directly targeting ERRα for ubiquitination and degradation.

**Fig. 3 ERRα phosphorylation by GSK3β leads to its proteasome-mediated degradation. a** Temporal profiles for insulin-induced phosphorylation changes of hepatic AKT, GSK3β and ERRα[22]. Data plotted are the median fold-changes, error bars denote SEM from at least three values, the exact sample size is in Source Data. **b** ERRα and phosphorylated ERRα protein levels in response to insulin stimulation. HepG2 cells were serum starved overnight prior to 0.58 μg/ml insulin treatment for 15 min. **c** ERRα and phosphorylated ERRα protein levels following GSK3β knockdown in HepG2 cells. **d** Schematic of ERRα protein structure and sequence alignment of putative GSK3β consensus phosphorylation motifs. Potential phosphorylated residues (S19, S22, S26) and priming site (E30) are indicated. **e** Representative in vitro kinase assay performed with HA-tagged GSK3β as well as in vitro-translated ERRα WT and phospho-mutant recombinant proteins including S19A, S22A, S26A, 3SA (S19A + S22A + S26A) and S26E. Phosphorylated and total ERRα proteins were detected by [32]P and coomassie brilliant blue, respectively. Experiments were independently repeated three times and quantified as shown on the right panel. Data are presented as means ± SEM (*p < 0.05, unpaired two-tailed Student's t test). **f** 293T cells were transfected with Flag-ERRα WT or 3SA mutant in the presence of HA-GSK3β WT or CA mutant for 48 h. **g** HepG2 cells stably overexpressing Flag-ERRα WT or 3SA mutant were treated with 10 μg/ml insulin for 30 min after overnight serum starvation. **h** Hepatic ERRα protein and ubiquitination levels in response to fasting and refeeding. Each lane represents immunoprecipitation performed using liver lysates prepared from two mice. **i** ERRα protein and ubiquitination levels in HEK293T cells stably expressing either control shRNA or 2 distinct shRNAs targeting GSK3β. Cells were transiently transfected with HA-Ub vector for 48 h and treated with MG132 before collection. **j** 293T cells were transfected with HA-GSK3β WT or CA mutant plasmids for 48 h and cells were collected after DMSO or MG132 treatment. **k** Proposed model of insulin-GSK3β-mediated ERRα stabilization. Insulin prevents ERRα from GSK3β-mediated phosphorylation, ubiquitylation, and subsequent degradation via the proteasome. Source data are provided as a Source Data file.

The E3 ligase FBXW7α is of particular interest as it mainly localizes in the nucleus and targets transcription factors such as PGC1α, c-Myc, SREBP1α and REV-ERBα[26,27]. Of note, FBXW7 specifically targets phosphorylated substrates bearing the consensus sequence (T/S-P-X-X-S/T/D/E) known as Cdc4-phosphodegron (CPD)[28]. We identified two putative CPD motifs in the amino terminus of ERRα surrounding residues S19, S22 and S26 (Fig. 4b). Remarkably, these residues are the GSK3β-targeted sites that we have defined above, suggesting that this composite sequence is cooperatively regulated by GSK3β and FBXW7. Using hepatocytes, we confirmed that endogenous FBXW7 and ERRα physiologically interact with each other (Fig. 4c). We next examined whether phosphorylation on the CPD motifs is required for ERRα levels to be controlled by FBXW7α. Co-IP and reciprocal co-IP showed that the phospho-defective CPD mutations largely blunted the interaction between FBXW7α and ERRα (Fig. 4d), while phosphorylation of the CPD motif did not affect ERRα interaction with other E3 ligases we uncovered above or previously published (Supplementary Fig. 4h). Consequently, FBXW7α promoted the ubiquitination (Fig. 4e) and subsequent degradation (Fig. 4f) of WT ERRα but not the CPD-disrupted ERRα 3SA mutant. Also, overexpression of FBXW7 greatly accelerated the degradation of WT ERRα without having any influence on the ERRα 3SA mutant (Fig. 4g). These data establishes that ERRα phosphorylation at S19, S22 and S26 is crucial for its recognition and ubiquitination by FBXW7 and subsequent degradation via the proteasome. Moreover, knock-down of GSK3β abolished the ERRα-FBXW7 interaction, while overexpression of the constitutively active GSK3β mutant enhanced this interaction (Fig. 4h, i). Accordingly, genetic, pharmacological, and physiological inhibitions of GSK3β were all proficient in blocking FBXW7-mediated ERRα degradation (Fig. 4j, Supplementary Fig. 4i, j). In vivo, we observed ERRα accumulation in acute (Supplementary Fig. 4k) and long-term FBXW7-depleted livers (Fig. 4k and Supplementary Fig. 4l). Also, the accumulated ERRα could not be further induced by refeeding in the absence of FBXW7 (Fig. 4k). Changes of ERRα protein levels in the FBXW7-null livers were independent of transcriptional regulation (Fig. 4l). Taken together, our results reveal that the E3 ligase FBXW7 is essential for the control of ERRα stability downstream of the insulin/GSK3β axis.

**ERRα[3SA] mice display transcriptional reprogramming and compromised metabolic homeostasis.** To explore the patho-physiological impact of disturbing the insulin/GSK3β/FBXW7/ERRα axis, we specifically impeded this pathway by genetically mutating the GSK3β/FBXW7 co-targeted ERRα serine residues S19, S22 and S26 to alanine in vivo using a CRISPR/Cas9 approach (Fig. 5a). Concurrently with ERRα[3SA] (S19A + S22A + S26A) engineered mice, we also generated ERRα[2SA] (S19A + S22A) and the single ERRα[S22A] phospho-mutant mice, all of which were viable and fertile. These point mutations were validated by Sanger sequencing, PCR genotyping, and use of a S19-targeted phospho-ERRα (S19) antibody[18] (Supplementary Fig. 5a–c). In vivo abrogation of ERRα phosphorylation at these sites resulted in its stabilization with the receptor being most accumulated in the ERRα[3SA] liver in comparison with ERRα[2SA] and ERRα[S22A] (Supplementary Fig. 5c), indicating that complete inactivation of both CPD motifs maximally protects ERRα from UPS-mediated degradation. Thus, we proceeded to fully characterize the ERRα[3SA] model. ERRα[3SA] mutagenesis significantly attenuated hepatic ERRα ubiquitination in vivo (Supplementary Fig. 5d). In parallel, we observed that ERRα[3SA] protein was similarly elevated in skeletal muscle (Fig. 5b), another major site of insulin action/resistance. Consistently, ERRα protein but not mRNA was stabilized in C2C12 myoblast cells upon proteasome inhibition (Supplementary Fig. 5e, f) and the functionally dominant FBXW7 isoform[29] Fbxw7a displayed similar expression levels in liver and muscle (Supplementary Fig. 5g, h), indicating that the functionality of the insulin-GSK3β-FBXW7-ERRα axis is conserved in insulin-sensitive tissues. Indeed, ERRα was simultaneously induced by refeeding in both liver and muscle, which was abrogated in ERRα[3SA] mice (Fig. 5c), signifying that the GSK3β-defective ERRα[3SA] is no longer regulated by the insulin signaling cascade in vivo.

To investigate the impact of the insulin unresponsive ERRα[3SA] on gene expression, we performed RNA sequencing of both mouse liver and muscle obtained from WT and ERRα[3SA] littermates, which revealed 764 DEGs in liver and 884 DEGs in muscle (Fig. 5d). Over 20% of DEGs identified in ERRα[3SA] mice were regulated by insulin, underscoring once again the critical role of ERRα in insulin action. Pathway analysis of ERRα[3SA] DEGs co-regulated by insulin revealed unique biological processes manifested in a tissue-specific manner. The top-enriched pathways deregulated in ERRα[3SA] liver were annotated to viral defense and interferon signaling, while mitochondrial-related processes including OXPHOS and mitochondrion organization, as well as lipid catabolism were highly enriched and upregulated in ERRα[3SA] muscle (Fig. 5d, Supplementary Fig. 5i).

ND-fed ERRα[3SA] mice tended to be slightly heavier than their littermate controls, although the data did not reach statistical significance (Fig. 5e, Supplementary Fig. 5j). Examination of animals in the DIO setting by feeding them a long-term HFD

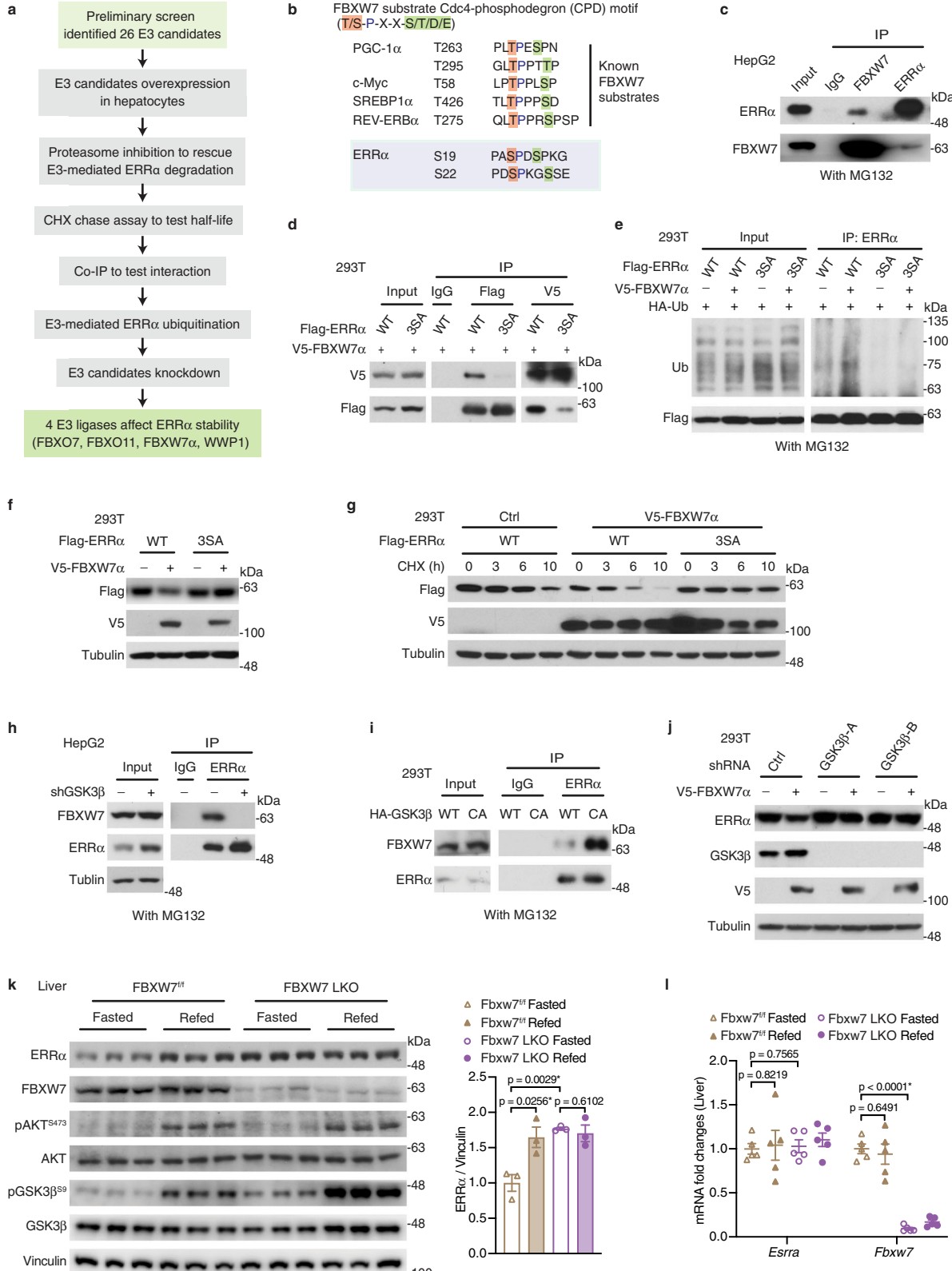

showed that ERRα³ˢᴬ mice moderately gained more weight contributed by increased fat mass (Fig. 5f, g). Both ND- and HFD-fed ERRα³ˢᴬ mice displayed modest but observably higher liver fat contents (Fig. 5h, Supplementary Fig. 5k) without changes in serum AST levels (Supplementary Fig. 5l). White adipocytes from ND-fed ERRα³ˢᴬ mice were slightly larger than that of WT, while HFD-fed ERRα³ˢᴬ mice displayed a tendency toward smaller white adipocytes (Fig. 5h, Supplementary Fig. 5m). This alternation implies increased adipose lipolysis in ERRα³ˢᴬ mice under a HFD, consistent with the notion that impaired insulin action increases adipose tissue lipolysis and promotes ectopic lipid re-esterification.

To gain further insight into the global effects of ERRα phospho-mutation on whole-body energy metabolism, metabolic

**Fig. 4 GSK3β-mediated ERRα degradation is dependent on FBXW7. a** Devised stratagem to identify and validate E3 ligase(s) targeting ERRα. See also Supplementary Fig. 4a. **b** High-affinity CPD motifs presented in previously established FBXW7 substrates including PGC-1α, c-Myc, SREBP1α, and REV-ERBα. Schematic of the two putative CPD motifs identified in ERRα spanning from residues serine 19 to serine 26, phosphorylated sites are highlighted **c** Co-IP of endogenous ERRα and FBXW7 in HepG2 cells treated with MG132. **d** Co-IP and reciprocal co-IP were performed using lysates from 293T cells transiently co-expressing V5-FBXW7 and Flag-ERRα WT or 3SA mutant. **e** Ubiquitination levels of ERRα WT or 3SA mutant in HEK293T cells transiently co-expressing the indicated constructs, cells were treated with MG132 prior to collection. **f** Protein levels of ERRα WT or 3SA in the presence or absence of FBXW7 in HEK293T cells. **g** HEK293T cells were co-transfected with the indicated constructs for 36 h prior to treatment with protein synthesis inhibitor cycloheximide (CHX, 100 µg/ml) for the indicated times. **h** Co-IP of endogenous ERRα and FBXW7 in GSK3β-silenced HepG2 cells treated with MG132. **i** Co-IP of FBXW7 and ERRα in HEK293T cells transiently transfected with HA-GSK3β WT or CA mutant. Cells were treated with MG132 prior to collection. **j** V5-FBXW7 and empty vectors were transiently transfected into HEK293T cells stably expressing either control or shRNAs targeting GSK3β. **k** Immunoblots (left, each lane represents liver from one mouse, n = 3 per group) and quantification (right) of ERRα protein levels in control and FBXW7-null livers during the fasting-refeeding transition. **l** *Esrra* and *Fbxw7* mRNA levels in control and FBXW7-null livers upon a fasting-refeeding transition (n = 5 per group). Data are presented as means ± SEM, *p < 0.05, unpaired two-tailed Student's t test (**k**, **l**). Source data are provided as a Source Data file.

cages were used for the simultaneous measurement of food intake, physical activity, energy expenditure and respiratory exchange ratio (RER) in ERRα[3SA] mice and their littermate controls. ND-fed ERRα[3SA] mice exhibited significantly lower resting energy expenditure (Fig. 5i, j) independent of changes in food intake or locomotor activity (Supplementary Fig. 5n), partly explaining their increased body weight trend. The RER, an indicator of carbohydrate-to-fat oxidation, derived by dividing the level of carbon dioxide produced by oxygen consumed, was found similar between genotypes upon a ND (Fig. 5k, Supplementary Fig. 5o). However, after long-term high-fat feeding, ERRα[3SA] mice displayed significantly higher RER (Fig. 5l) without obvious alterations in other metabolic parameters (Supplementary Fig. 5p). These data show that ERRα[3SA] mice rely more heavily on carbohydrates for energy production in response to a HFD, likely caused by their impaired fat utilization capacity. Together, our results reveal that ERRα[3SA] mice exhibit impaired metabolic homeostasis under either a ND or HFD, indicating their increased risk of developing metabolic disorders.

**ERRα[3SA] phospho-deficient mice develop insulin resistance.** Impaired glucose regulation is considered a necessary stage and an early warning signal of insulin resistance. Under a ND, ERRα[3SA] mice displayed markedly elevated blood glucose levels without differences in blood lactate concentrations (Fig. 6a, Supplementary Fig. 6a, b). Hyperinsulinemia accompanied the hyperglycemia observed in ERRα[3SA] mice (Fig. 6b). While fasting (24 h) glucose and insulin levels were similar between genotypes (Fig. 6a, b), a switch from fasting to refeeding led to significantly higher postprandial blood glucose levels in ERRα[3SA] mice despite a similar rise of serum insulin concentration relative to WT littermates (Fig. 6a, b). This observed reduction in insulin sensitivity of ERRα[3SA] mice was further examined at a molecular level by investigating the insulin signaling pathway in response to refeeding. Insulin-induced AKT phosphorylation of S473 and T308 were significantly impaired in ERRα[3SA] liver, in association with reduced GSK3β phosphorylation and GS dephosphorylation (Fig. 6c, Supplementary Fig. 6c). In skeletal muscle, ERRα[3SA] mice showed only a modest impairment in insulin-stimulated AKT T308 phosphorylation and no difference in AKT S473 phosphorylation compared to WT littermates (Fig. 6c, Supplementary Fig. 6c). Nonetheless, the effects of insulin on GSK3β and GS phosphorylation were mostly blunted in ERRα[3SA] muscle, accounting for the parallel defect in postprandial glycogenesis in muscle (Fig. 6c, d, and Supplementary Fig. 6c). No significant differences in hepatic glycogen levels were observed between genotypes under a ND (Fig. 6d). In response to a long-term HFD, ERRα[3SA] mice presented reduced hepatic glycogen contents (Fig. 6e, Supplementary Fig. 6d) and fasting hyperglycemia (Fig. 6f), indicating increased hepatic glucose production. Lipid

metabolism was also dysregulated in HFD-fed ERRα[3SA] mice as reflected by elevated serum triglyceride (TG) and free fatty acid (FFA) levels (Fig. 6g, h) while no significant differences were observed in ND-fed animals (Supplementary Fig. 6e). Dysregulated circulating TG and FFA are strongly correlated with insulin resistance, partly by reducing whole-body insulin-stimulated glucose disposal[2,30]. Consistently, HFD-fed ERRα[3SA] mice displayed elevated blood glucose concentrations after exogenous glucose loading and insulin stimulation during a glucose tolerance test (GTT) and an insulin tolerance test (ITT), respectively (Fig. 6i, j), although only slight differences were observed between ND-fed animals (Supplementary Fig. 6f, g).

We next aimed to explore the genetic causes driving the insulin resistance in ERRα[3SA] mice. Remarkably, *Pdk4*, a classic ERRα target[31], was noted in a screen for insulin-regulated genes that were simultaneously altered in ERRα[3SA] liver and muscle (Fig. 6k). PDK4 represses glucose oxidation by directly inhibiting the pyruvate dehydrogenase (PDH) complex, which is the rate-limiting enzyme controlling the flux of pyruvate into mitochondria. *Pdk4* is upregulated under pathological conditions, such as HFD-induced insulin resistance and T2D[32,33]. ERRα[3SA] mice displayed elevated *Pdk4* levels to a similar extent to that of genetically obese diabetic mice and HFD-fed mice (Fig. 6l, Supplementary Fig. 6h, i), in line with increased ERRα[3SA] binding to the *Pdk4* promoter (Fig. 6m). ERRα[3SA] mice displayed reduced muscle PDH activity (Supplementary Fig. 6j), implicating decreased glucose oxidation. Moreover, liver and muscle tissues from mice treated with the GSK3β inhibitor SB216763 as well as FBXW7-null liver showed significantly elevated *Pdk4* mRNA levels (Supplementary Fig. 6k, l), indicating that the GSK3β-FBXW7-ERRα axis regulates glucose homeostasis at least partly via modulation of *Pdk4* expression. ND-fed ERRα[3SA] mice also showed dysregulated genes encoding proteins involved in hepatic lipid metabolism such as *Acot11* and *Angptl8* (Fig. 6n), which have been shown to be closely associated with insulin resistance and NAFLD[34,35]. In addition, inflammatory marker *Gdf15*[36], oxidative stress marker *Hmox1*[37], interferon-related genes (*Ifit1, Irf7, Oasl1, Rsad2*), tumor necrosis factor-related genes (*Tnfsf10, Tnfrsf11a*), and inflammation-related genes (*Irgm1, Tlr3*) were significantly elevated in ERRα[3SA] liver (Fig. 6n). Notably, these genes were mostly upregulated during DIO (Supplementary Fig. 6m), suggesting their contribution to the increased fat content in ND-fed ERRα[3SA] liver. Alternatively, we found pronounced downregulation of important genes associated with glucose metabolism in ERRα[3SA] muscle, such as genes encoding insulin receptor substrates (*Irs2, Gab1, Shc2*), glucose transporter 3 (*Slc2a3*), glucokinase (*Gck*), and AMPKg3 subunit (*Prkag3*), a predominant isoform in glycolytic skeletal muscle (Fig. 6o). Furthermore, genes encoding proteins involved in fatty acid utilization were upregulated in ERRα[3SA] muscle, including the

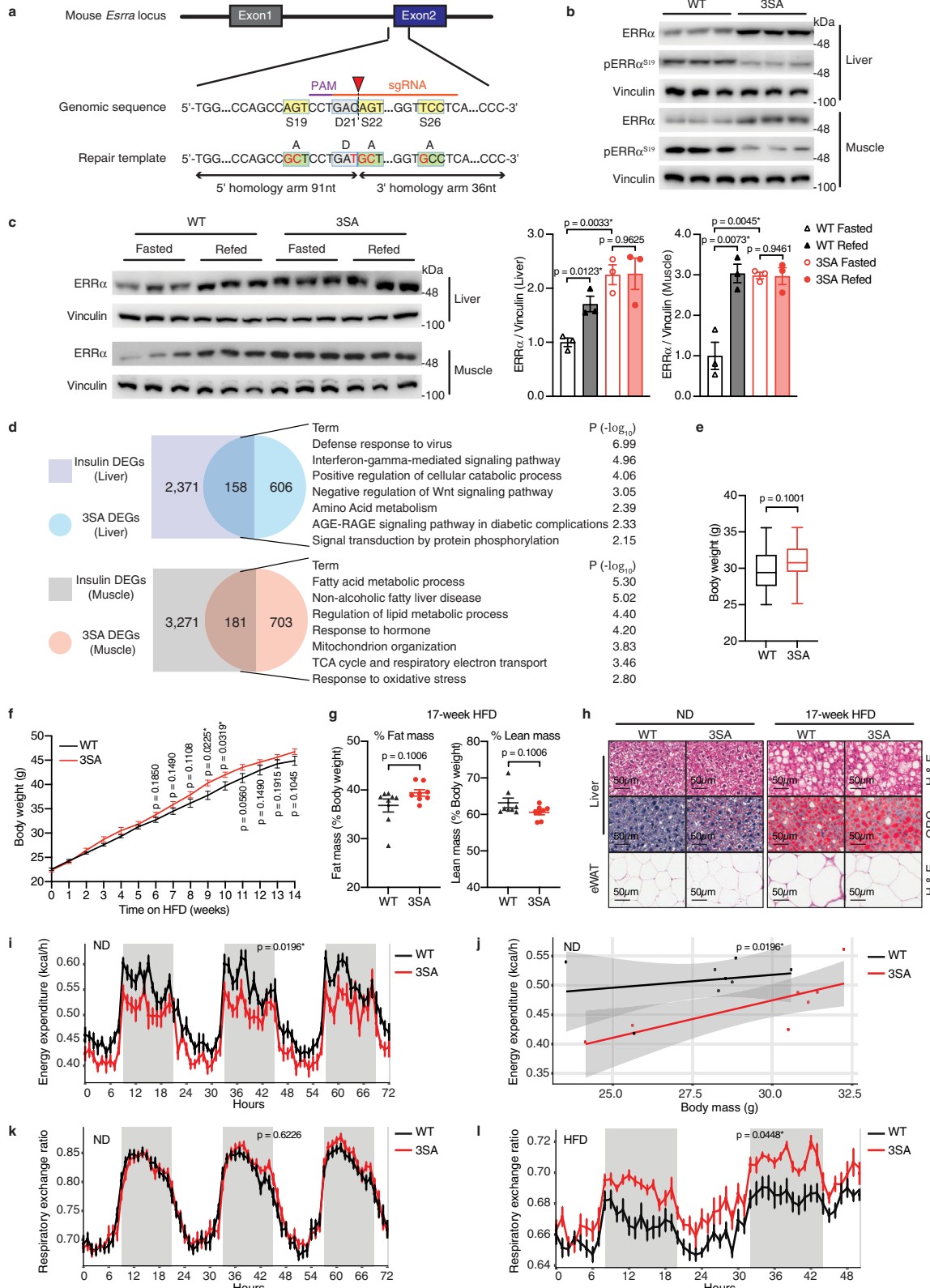

major skeletal muscle lipid transporter fatty acid-binding protein 3 (*Fabp3*), and lipoprotein lipase (*Lpl*) (Fig. 6o), a rate-limiting enzyme activated by insulin to stimulate TG hydrolysis and peripheral fatty acid (FA) uptake. Consistently, ERRα^3SA mice showed increased postprandial muscle FFAs (Fig. 6p), likely a compensation for impaired insulin-stimulated glucose uptake. Intramyocellular accumulation of saturated FFAs, lipid and lipid

metabolites have been shown to interfere with insulin signaling and provoke insulin resistance[38]. Thus, FA oversupply to ERRα^3SA skeletal muscle, especially upon a HFD, might further diminish glucose disposal in muscle, resulting in increased glucose diversion to liver. Indeed, HFD-fed ERRα^3SA mice displayed increased hepatic glucose metabolism, reflected by elevated expression of genes encoding glucokinase (*Gck*) and

**Fig. 5 ERRα3SA phospho-deficient mice display transcriptional reprogramming and compromised metabolic homeostasis. a** Schematic of the editing strategy for ERRα3SA mice. Cas9 cleavage site is indicated by the red arrowhead, the single-guide RNA (sgRNA) target sequence and the protospacer adjacent motif (PAM) are marked. Homology repair template contains desired mutations (red) with the resultant residue substitutions denoted. **b** Total and S19-phosphorylated ERRα proteins. Each lane represents tissue from one mouse, $n = 3$ per group. **c** Immunoblot (left, each lane represents tissue from one mouse, $n = 3$ per group) and quantification (right) of ERRα protein levels from WT and ERRα3SA littermates upon a fasting-refeeding transition. **d** Overlap and pathway analysis of DEGs identified in ND-fed ERRα3SA mice with insulin-regulated DEGs (GSE117741[62]) ($p < 0.05$, |FC| ≥ 1.20). See also Supplementary Fig. 5i. **e** Body weight of WT and ERRα3SA littermates on a ND (WT, $n = 41$; 3SA, $n = 45$). Box plots show centre line as median, bounds of box as 25th and 75th percentiles, whiskers as minima and maxima. **f** Body weights during a 14-week HFD feeding (WT, $n = 12$; 3SA, $n = 13$). **g** Percentage of fat and lean mass ($n = 8$ per group). **h** Representative images of liver hematoxylin and eosin (H&E) and oil red O (ORO) staining, and epididymal WAT (eWAT) H&E staining. Scale bars, 50 μm. See Supplementary Fig. 5k, m for quantification. **i** Whole-body energy expenditure and **j** regression plot of averaged energy expenditure vs. total body mass, and **k** respiratory exchange ratio (WT, $n = 8$; 3SA, $n = 7$). **l** Respiratory exchange ratio of WT and ERRα3SA littermates maintained on a 14-week HFD ($n = 7$ per group). Data are represented as means ± SEM (**c, f, g, i, k, l**). A generalized linear model is fitted to the data (**j**) and estimations are displayed by the line, with the shaded region representing the standard error. *$p < 0.05$, unpaired two-tailed Student's $t$ test (**c, e–g**), ANCOVA (**i, j**), and ANOVA (**k, l**). Source data are provided as a Source Data file.

---

pyruvate kinase (*Pklr*), two irreversible and thus rate-limiting glycolytic enzymes (Fig. 6q). Meanwhile, key lipogenic genes including *Cs, Acly, Acaca, Fasn, Elovl6, Scd1* and *Gpam* were all significantly upregulated in HFD-fed ERRα3SA liver without any changes in the expression of *Srebf1/2*, genes encoding the major transcription factors involved in lipid biosynthesis (Fig. 6q). In addition to promoting hepatic de novo lipogenesis (DNL), increased glycolysis in ERRα3SA liver could concurrently inhibit fatty acid oxidation (FAO), albeit no differences in FAO genes were observed between genotypes (Supplementary Fig. 6n). Additionally, genes encoding proteins involved in hepatic FA uptake (*Cd36, Got2*) and VLDL secretion (*Apob, Mttp, Cideb*) were elevated in HFD-fed ERRα3SA mice (Fig. 6q), consistent with their elevated liver and serum TG contents. Of note, increased glucose flux into hepatocytes did not lead to incorporation into liver glycogen, instead, the glycogenolysis genes *Pygl* and *Pgm1* were elevated in ERRα3SA liver upon a HFD (Fig. 6q), contributing to increased hepatic glucose production. Hepatic gluconeogenesis genes showed a mixed pattern of changes in HFD-fed ERRα3SA mice, with *Pck1* significantly decreased, and *Pcx* increased without any differences in *G6pc, Fbp1* and *Gpt* (Supplementary Fig. 6o). Overall, our results illustrate that unregulated ERRα activity in the ERRα3SA mice leads to postprandial hyperglycemia partly caused by impaired insulin-stimulated glucose disposal as well as increased hepatic inflammation and oxidative stress. Upon a HFD, increased FA intake and muscle insulin resistance leads to elevated hepatic glucose flux and transcriptional activation of glycolysis, DNL, FA uptake, VLDL-secretion as well as glycogenolysis, further exacerbating insulin resistance.

**Pharmacological ERRα inhibition improves metabolism in ERRα3SA mice.** Our studies have characterized ERRα3SA as a mouse model exhibiting excess ERRα protein and insulin resistance. To further validate that these metabolic defects in ERRα3SA mice are directly caused by ERRα overactivation, we treated these mice with C29, a specific ERRα inverse agonist[39]. In the ND-fed ERRα3SA mice, C29 ameliorated hyperglycemia without significant alterations of other parameters (Supplementary Fig. 7a–c). ERRα3SA mice were then fed a HFD for 6 weeks to exacerbate their metabolic disorders prior to 3-week intraperitoneal injections of vehicle or C29 at Zeitgeber time (ZT) 14 (Fig. 7a). C29-treated mice gradually lost body mass despite the HFD (Fig. 7b), which was mainly caused by reduced peripheral fat deposits (Supplementary Fig. 7d). Administration of 3 weeks of C29 in HFD-fed ERRα3SA mice dramatically alleviated hyperglycemia, hyperinsulinemia and hypertriglyceridemia without any changes in serum AST levels (Fig. 7c–f), indicating no liver toxicity. C29 also significantly enhanced glucose

clearance capacity in HFD-fed ERRα3SA mice upon exogenous glucose loading and insulin stimulation (Fig. 7g, h), indicating improved whole-body insulin sensitivity. This was further reflected molecularly by increased insulin-stimulated AKT phosphorylation at T308 and S473 in liver and skeletal muscle upon refeeding (Fig. 7i, j and Supplementary Fig. 7e), consistent with our in vitro observations (Supplementary Fig. 7f). To gain more insight into C29-induced weight loss and reversal of insulin resistance, HFD-fed ERRα3SA mice treated with vehicle or C29 were examined in metabolic cages (Fig. 7i). We found the metabolic benefits of C29 were partly caused by augmented fat utilization by C29, reflected by decreased RER in the dark phase (Fig. 7k), rather than changes in energy expenditure (Supplementary Fig. 7g). Food intake was also decreased in C29-treated mice, without differences in physical activity (Supplementary Fig. 7h, i).

We next fed ERRα3SA mice a long-term HFD for 18 weeks to induce NAFLD (Fig. 7l). We confirmed that C29 significantly reduced ERRα protein in ERRα3SA liver and muscle (Fig. 7m). Consistently, C29 injections in ERRα3SA mice lead to weight loss and improved glucose homeostasis (Supplementary Fig. 7j–l). In addition, C29 ameliorated HFD-induced fat accumulation in liver and adipocytes (Fig. 7n, Supplementary Fig. 7m, n). In parallel, C29 increased hepatic glycogen levels, which was independent of changes in liver glycogen synthase (GS) activity (Fig. 7n, o and Supplementary Fig. 7o). Alternatively, muscle GS phosphorylation was moderately reduced by C29 (Fig. 7o). To further determine the molecular mechanisms underlying the benefits of C29 in hyperactivated ERRα3SA mice, we performed gene expression analysis and found that C29 simultaneously diminished the elevated *Pdk4* levels in liver and muscle of these mice (Fig. 7p). Muscle *Irs2, Slc2a3*, and *Ugp2* were significantly upregulated by C29 (Fig. 7q), underpinning increased muscle glucose uptake and glycogenesis. C29-mediated reduction of circulating glucose was also partly contributed by decreased hepatic glucose production, reflected by downregulated hepatic glycogenolysis genes (*Pygl, Pgm1*) and gluconeogenesis genes (*Fbp1, Gpt*) (Fig. 7r, Supplementary Fig. 7p). Meanwhile, hepatic glycolytic genes (*Slc2a2, Gck, Pklr*) were significantly downregulated by C29 (Fig. 7r), in line with decreased circulating lactate (Supplementary Fig. 7l). Although C29 showed no impact on lipogenic genes (Supplementary Fig. 7p), C29-mediated inhibition of genes encoding proteins involved in glycolysis and FA uptake (*Got2*) would diminish precursors for DNL (Fig. 7r). Furthermore, C29 enhanced hepatic FAO genes (*Acox1, Ehhadh, Ech1, Acaa1a*) and downregulated genes encoding proteins involved in VLDL secretion (*Apob, Mttp, Pla2g12b, Cideb*), underpinning decreases in liver and serum TG levels (Fig. 7r). In addition, C29 significantly reversed the elevated expression of

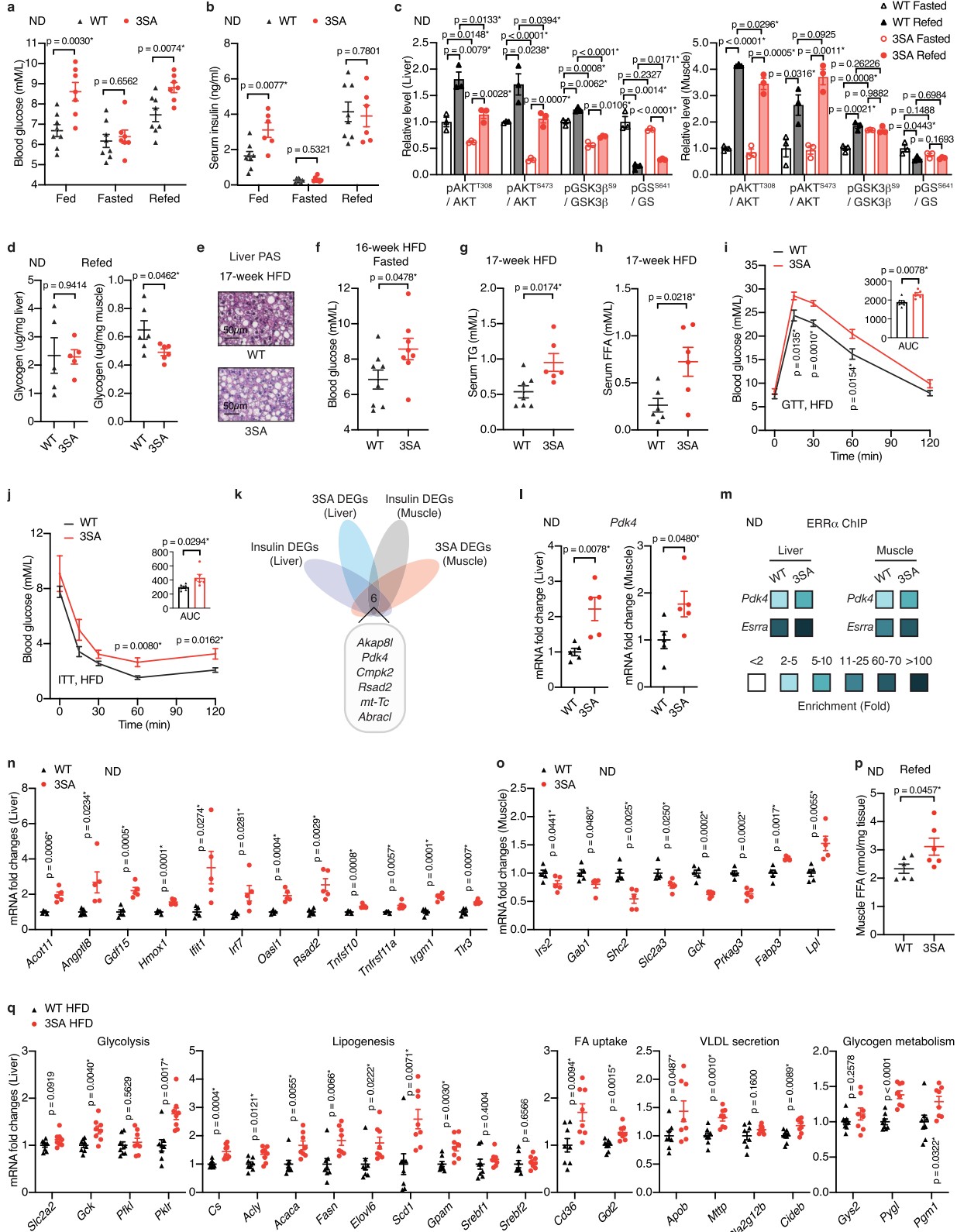

lipid-related genes (*Acot11*, *Angptl8*) and inflammation-related genes (*Gdf15*, *Hmox1*, *Rsad2*, *Tnfsf10*, *Tnfrsf11a*, *Irgm1*, *Tlr3*) in ERRα³ᵁᴬ liver (Supplementary Fig. 7p). Together, our results demonstrate that C29-mediated suppression of the stable and activated ERRα³ᴬ protein enhances whole-body insulin sensitivity as well as provides anti-obesity, anti-hyperglycemia/insulinemia/triglyceridemia, and anti-hepatic steatosis effects in

HFD-fed ERRα³ᴬ mice. Mechanistically, C29 enhances skeletal muscle glucose oxidation while inhibiting hepatic glucose production. Simultaneously, C29 inhibits hepatic glycolysis and FA uptake, limiting the precursors for TG synthesis. C29 further promotes hepatic TG clearance by augmenting FAO and concomitantly decreases circulating TG by suppressing VLDL secretion.

**Fig. 6 ERRα[3SA] mice develop insulin resistance. a** Blood glucose concentrations (WT, $n = 8$; 3SA, $n = 7$), **b** serum insulin levels of ND-fed ERRα[3SA] and WT littermates in the fed (WT, $n = 8$; 3SA, $n = 6$, fasted ($n = 7$ per group), and refer (WT, $n = 8$; 3SA, $n = 6$) conditions. ($n = 8$; 3SA, $n = 6$; WT fasted, $n = 7$; 3SA fasted, $n = 7$; WT Refed, $n = 8$; 3SA Refed, $n = 6$), **c** Quantification of the insulin signaling cascade components from ND-fed WT and ERRα[3SA] littermates upon fasting and refeeding ($n = 3$ per group). See Supplementary Fig. 6c for immunoblot image. **d** Liver ($n = 5$ per group) and muscle ($n = 6$ per group) glycogen levels of ND-fed ERRα[3SA] and WT littermates upon refeeding. **e** Representative PAS staining of ERRα[3SA] and WT littermates post 17-week high-fat feeding. Scale bars, 50 μm. See also Supplementary Fig. 6d for quantitation. **f** Fasting blood glucose concentrations upon a 16-week HFD ($n = 8$ per group). **g, h** Concentrations of fed serum TG (**g**) (WT, $n = 7$; 3SA, $n = 6$) and FFA (**h**) ($n = 6$ per group) of ERRα[3SA] and WT littermates after 17-week high-fat feeding. **i** Tolerance test of glucose (GTT) and **j** insulin (ITT) performed after 6-week and 8-week HFD, respectively. Inset graphs show area under the curve (AUC) ($n = 6$ per group). **k** Overlap of DEGs identified in ND-fed ERRα[3SA] mice with insulin-regulated DEGs (GSE117741[62]) ($p < 0.05$, |FC| ≥ 1.20). **l** Pdk4 mRNA levels ($n = 5$ per group). **m** ChIP-qPCR analysis of ERRα binding to Esrra or Pdk4 promoters. **n, o** mRNA levels of the indicated genes upon a ND ($n = 5$ per group). **p** Muscle FFA levels upon refeeding ($n = 6$ per group). **q** mRNA levels of genes encoding proteins involved in glucose and fat metabolism in liver from ERRα[3SA] and WT littermates fed a HFD for 17 weeks ($n = 8$ per group). Data are presented as mean ± SEM, *$p < 0.05$, unpaired two-tailed Student's $t$ test (**a–d**, **f–j**, **l**, **n–q**). Source data are provided as a Source Data file.

## Discussion

Herein, using a combination of several mouse models and molecular tools, we uncovered an insulin-GSK3β-FBXW7-ERRα signaling axis that fulfills a critical gap in our understanding of the transcriptional readout of insulin action and the crucial role played by the nuclear receptor ERRα in insulin signaling. We found that insulin leads to modulation of the ERRα phosphorylation state via GSK3β, which directs its recognition by FBXW7 thus governing its ensuing ubiquitination and overall stability. Importantly, abrogation of the GSK3β-mediated ERRα phosphorylation at S19, S22 and S26 in vivo led to altered insulin-dependent liver and muscle transcriptomes, compromised metabolism and insulin resistance (Supplementary Fig. 8).

In addition to FBXW7 and GSK3β, our study revealed that E3 ligases FBXO11, FBXO7, WWP1 and kinase CK2 can also regulate ERRα protein stability. The extent and physiological relevance of their associations with ERRα remain to be defined. Further, we observed that ERRα is also activated by insulin, resulting in similar ERRα accumulation. It is currently unknown which E3 ligases participate in the underlying mechanisms. Likewise, the phosphatase(s) involved in ERRα dephosphorylation remains elusive. Identification of an ERRα phosphatase, if required, remains to be explored. Considering that PP2A is known to dephosphorylate GSK3β[40], it might be of interest to investigate whether it is involved in the insulin/GSK3β/BXW7/ERRα axis.

The ERRα[3SA] and ERRα-null mice displayed opposite metabolic phenotypes in many aspects, indicating these phenomena depend primarily on the relative abundance and activity of ERRα protein. First, ERRα phospho-mutagenesis resulted in impaired glucose homeostasis. ERRα has been implicated in the inhibition of skeletal muscle glucose catabolism by transcriptional activation of Pdk4, a negative regulator of glucose oxidation that is elevated in diabetes[31,32]. Patients with obesity and T2D display "metabolic inflexibility" manifested by impaired insulin-stimulated skeletal muscle glucose oxidation[33,41]. We observed increased Pdk4 levels in ERRα[3SA] mice together with elevated ERRα binding to the Pdk4 promoter uncovering a prominent regulatory role for ERRα in metabolic flexibility. In addition to suppressed glucose oxidation, attenuated insulin signaling and increased FA flux also contributed to muscle insulin resistance and reduced postprandial muscle glycogenesis in ERRα[3SA] mice. It will be interesting to explore whether ERRα is responsible for insulin-stimulated glucose disposal in skeletal muscle that are independent of the PI3K-AKT signaling pathway, as attenuation of refeeding-induced AKT phosphorylation is minimal in ERRα[3SA] muscle, and multiple studies revealed additional pathways perturbing muscle glucose tolerance and insulin sensitivity in addition to attenuated AKT signaling[3,42,43]. ERRα has also been shown to negatively regulate hepatic gluconeogenesis by repressing Pck1[44], and insulin-mediated repression of hepatic Pck1 was abolished in ERRα-

null mice, demonstrating that ERRα represses Pck1 in the physiological fed state. We observed downregulated Pck1 expression in HFD-fed ERRα[3SA] liver despite their hyperglycemic phenotype. On the other hand, HFD-fed ERRα KO mice displayed fasting hypoglycemia[12]. Thus, our results show that gluconeogenesis is not the central cause of the observed glycemic phenotypes in theses contexts. Conversely, hepatic glycogenolysis was found upregulated in HFD-fed ERRα[3SA] mice, contributing to their elevated fasting blood glucose levels. Second, ERRα[3SA] mice exhibited aberrant lipid metabolism. Diminished muscle glucose disposal in HFD-fed ERRα[3SA] mice was compensated by increased hepatic glucose uptake and metabolism, which was used for lipid synthesis rather than glycogenesis, suggesting hyperglycemia in diabetes is unlikely due to decreased hepatic glycolysis. Indeed, we observed elevated RER in HFD-fed ERRα[3SA] mice together with upregulated hepatic glycolysis and lipogenesis. Further, hepatic VLDL secretion, which was blunted in ERRα LKO mice[45], was favored in HFD-fed ERRα[3SA] mice and contributed to elevated circulating TG. Although modestly elevated hepatic fat content and hyperglycemia were observed in ERRα[3SA] mice under both a ND and HFD, essential genes in lipid and lipoprotein metabolism were only altered after a long-term HFD, reflecting an important gene-environment interaction. Third, ERRα[3SA] mice demonstrated enhanced skeletal muscle mitochondrial functions. Mitochondrial-related genes were exclusively upregulated in ERRα[3SA] muscle, although both liver and muscle displayed comparably elevated ERRα protein levels. This observed difference in these two insulin-responsive tissues is likely caused by the presence and relative abundance of specific transcription factor coregulators[46]. A large amount of literature suggests targeting skeletal muscle mitochondria to prevent or treat T2D. However, it is still unknown whether mitochondrial dysfunction is a consequence or a cause of insulin resistance, and the idea of boosting mitochondrial OXPHOS to treat diabetes is currently debated[47–49]. We observed that ERRα[3SA] mice developed insulin resistance despite elevated muscle mitochondrial OXPHOS gene expression, partly caused by their reduced PDH activity, which would limit the entrance of pyruvate into mitochondria. Also, ERRα KO mice with impaired mitochondria are protected from HFD-induced insulin resistance[12]. Thus, our studies suggest that insulin resistance is not necessarily caused by deficiency or enhancement of mitochondrial functions, probably because multiple factors can cause insulin resistance without the involvement of mitochondria. In agreement with our observations, it has been previously indicated that reduced/enhanced OXPHOS capacity is unlikely to be the central cause of all forms of insulin resistance, but mitochondrial dysfunction might exacerbate insulin resistance via increasing reactive oxygen species production[50]. Interestingly, ERRα has recently been shown to also act as a reactive oxygen species sensor[51]. Together, these clear

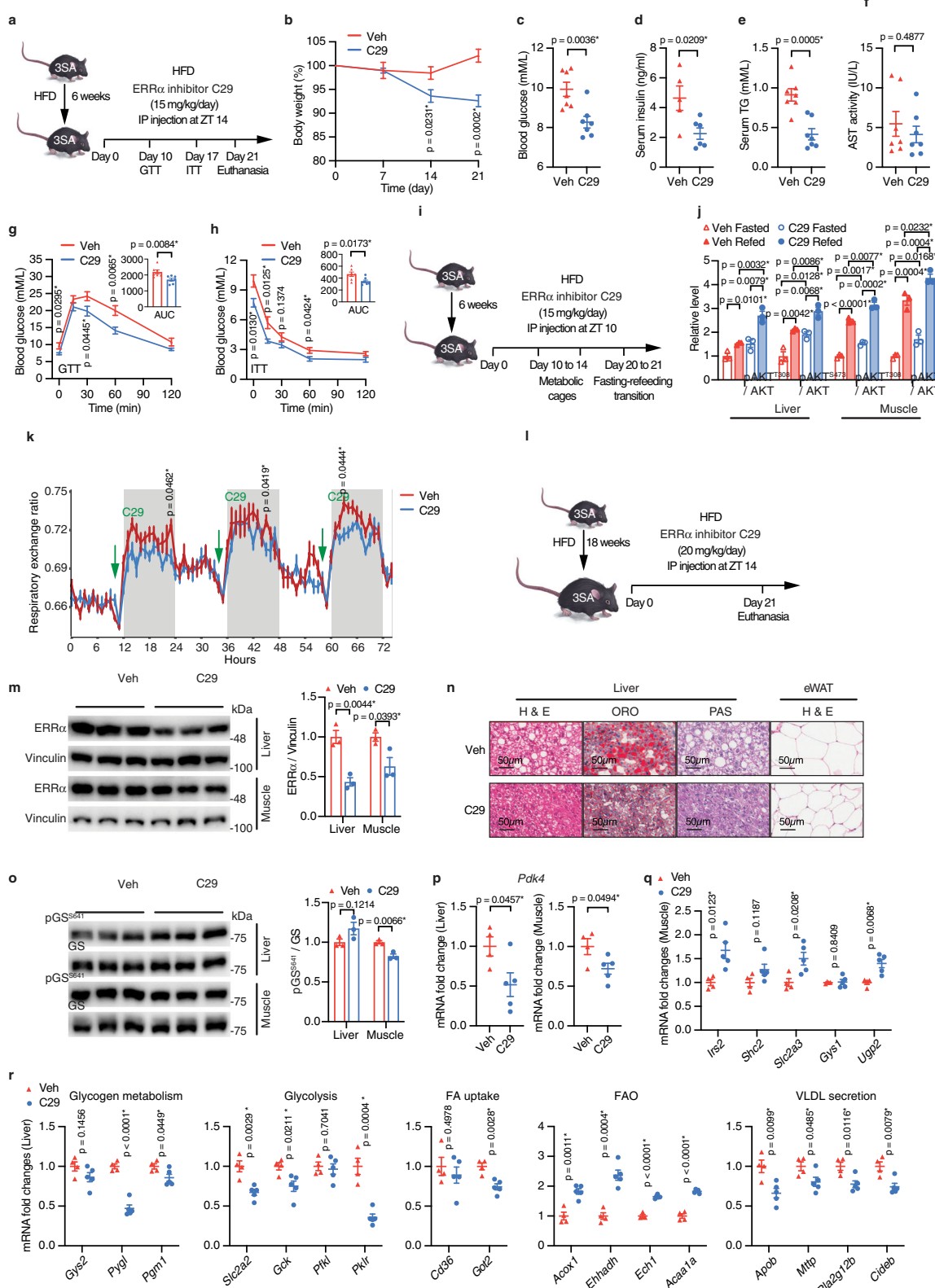

opposite phenotypes between ERRα[3SA] and KO mice provide a strong proof of concept that ERRα protein levels and activity directly correlate with insulin sensitivity. Critically, we observed that the pharmacological agent C29 (ERRα inverse agonist) ameliorated the HFD-induced NAFLD in the ERRα[3SA] mice concomitantly with the reversal of dyslipidemia and hyperglycemia, thereby increasing insulin responsiveness.

In conclusion, this work revealed that specific ERRα phosphorylation serves as a nutritional sensor and establish ERRα[3SA] mice as a valuable insulin resistance model, while further demonstrating the metabolic potency of ERRα-based pharmacotherapies. An important future direction will be to uncover the relationships between ERRα phosphorylation and human diseases for the development of alternative targeted therapeutics.

**Fig. 7 Pharmacological ERRα inhibition improves metabolic health of ERRα$^{3SA}$ mice. a–h** Effects of C29 (**b**, **c**, **e**, **f**, **g**, **h** $n = 7$ per group; **d** $n = 5$ for Veh, $n = 6$ for C29). **a** Schematic of C29 administration. Six-week-old ERRα$^{3SA}$ mice were fed a HFD for 6 weeks and then assigned to two groups and subjected to injection with vehicle or C29 before GTT and ITT. Mice were euthanized and examined after 21 days of injections. **b** Body weight changes (%). Level of fed blood glucose (**c**), serum insulin (**d**), TG (**e**) and AST activity (**f**) after 21 days of injections. **g** GTT and **h** ITT. Inset graphs show AUC. **i–k** Effects of C29 on insulin sensitivity and energy metabolism. **i** ERRα$^{3SA}$ mice fed a HFD for 6 weeks were subjected to injections with vehicle or C29 for 10 days before examination in the metabolic cages for 4 days. After 21 days of C29 treatment, mice were subjected to fasting and refeeding challenges. **j** Quantification of AKT phosphorylation upon fasting and refeeding ($n = 3$ per group). See Supplementary Fig. 7e for immunoblot image. **k** Effects of C29 on respiratory exchange ratio ($n = 7$ per group). **l–r** Effects of C29 on NAFLD (**m**, **o** $n = 3$ per group; **p**, **q**, **r** $n = 4$ for Veh and $n = 5$ for C29). **l** ERRα$^{3SA}$ mice fed a HFD for 18 weeks were subjected to injections with vehicle or C29 for 21 days before euthanasia and examination. **m** Immunoblots (left, each lane represents tissue from one mouse) and quantification (right) of ERRα protein levels. **n** Representative images of liver and eWAT sections stained with either H&E, ORO or PAS. Scale bars, 50 μm. See also Supplementary Fig. 7m–o for quantification. **o** Immunoblots (left, each lane represents tissue from one mouse) and quantification (right) of GS phosphorylation levels. **p–r** mRNA levels of the indicated genes. Data are represented as means ± SEM. *$p < 0.05$, unpaired two-tailed Student's $t$ test (**b–h**, **j**, **k**, **m**, **o–r**). Source data are provided as a Source Data file.

---

Considering that many diseases are linked to altered protein phosphorylation states[52], screening for drugs specifically targeting the insulin-ERRα signaling pathway is likely to confer significant therapeutic benefits.

## Methods
This study complies with all relevant ethical regulations of McGill University.

**Mice.** All mouse manipulations were performed in accordance with procedures approved by the McGill Facility Animal Care Committee within animal protocol 3173, and complied with ethical guidelines set by the Canadian Council of Animal Care. Unless otherwise specified, all experiments used age-matched male littermates (2- to 3-month-old). Mice were housed two to five per cage in a constant environment (ambient temperature: 18–24 °C; relative humidity: 30–70%) under a 12-h light/dark cycle (7am–7pm light, 7pm–7am dark) with ad libitum access to water and a standard normal diet (ND; Envigo; Teklad Rodent diet 2920x; 3.1 kal/g, 24 kcal% protein, 16 kcal% fat, 60 kcal% carbohydrate) in an animal facility at McGill University. Mice were euthanized by cervical dislocation at Zeitgeber time (ZT) 2–6 for serum and tissue isolations. Liver, skeletal muscle (gastrocnemius and soleus), and white adipose tissue were harvested and snap-frozen in liquid nitrogen and stored at −80 °C until processing.

FBXW7 floxed mice (Stock No: 017563)[53] and GSK3β floxed mice (Stock No: 029592)[54] on a C57BL/6J genetic background were obtained from the Jackson Laboratory and bred with Alb-Cre mice. Homozygous floxed and hemizygous Cre transgenic males were bred with homozygous floxed females to generate Cre-positive and Cre-negative experimental mice, see Supplementary Data 7 for genotyping primers. Male C57BL/6J mice (Stock No: 000664), B6.Cg-Lep$^{ob}$/J mice (Stock No: 000632)[55], and B6.BKS(D)-Lepr$^{db}$/J mice (Stock No: 000697)[56] aged 6 weeks were purchased from the Jackson Laboratory and allowed to acclimate for 2 weeks in our animal facility before tissue collection and examination.

ERRα phospho-mutant mice on a C57BL/6N genetic background were generated by CRISPR/Cas9 gene editing at the McGill Integrated Core for Animal Modeling (MICAM). Founders were sequenced using forward 5′-GCTTAGTC CCAGTGTTGCCC-3′ and reverse 5′- CTGACTCAAGCAGCAACAGAC-3′ primers to determine the ERRα point mutations. Transgenic lines were backcrossed with C57BL/6N mice directly purchased from Envigo to segregate alleles with potential off-targets or random mutations induced by CRISPR/Cas9. Resulting pups were sequenced as described above as well as genotyped with primers carrying specific mutations, see Supplementary Data 7 for primer sets used. Heterozygous transgenic mice were mated to generate homozygous ERRα WT and phospho-mutant littermates for experiments.

For the fasting and refeeding experiments, mice were transferred from cages with corn chip to wood chip bedding and were randomly assigned to fasted and refed groups. The fasted group was deprived of food for 24 h, and the refed group was fasted for 22 h followed by refeeding with a ND for 2 h.

For glucose and insulin stimulations, mice were intraperitoneally injected with saline, 2 g/kg D-glucose (cat. no. 15023-021; Thermo Fisher Scientific) or 1 unit/kg insulin (cat. no. I0516; Sigma) for 0.5 h after 23.5 h fasting. For liver RNA-sequencing of insulin stimulation, ERRα WT or KO mice on a C57BL/6N genetic background[13] were randomly assigned to two groups and fasted for 12 h overnight followed by intraperitoneal injection of saline or 1 unit/kg insulin. At 4 h post insulin stimulation (ZT6), mice were euthanized, and tissues were isolated.

For experiments involving high-fat diet (HFD) or chow diet (CD), mice were fed a HFD (cat. no. TD.06414; Envigo; Teklad Custom diet: 5.1 kal/g, 18.3 kcal% protein, 60.3 kcal% fat, 21.4 kcal% carbohydrate) or a CD (cat. no. TD.08806; Envigo; Teklad Custom diet: 3.6 kal/g, 20.5 kcal% protein, 10.5 kcal% fat, 69.1 kcal% carbohydrate) for the indicated time, initiated at 6 weeks of age.

For drug administration, ERRα inhibitor C29[39] and GSK3β inhibitor SB 216763 (cat. no. HY-12012; MedChemExpress) were delivered daily by intraperitoneal injections at the indicated time and duration. Drugs were first dissolved in DMSO

(Thermo Fisher Scientific) then diluted into 5 mg/ml with dilution buffer (30% PEG-300 (Sigma), 70% saline). Vehicle solution was prepared by diluting equal amount of DMSO with dilution buffer.

**Cell culture.** HepG2, HEK293T, and C2C12 cells were from the ATCC and cultured in DMEM (cat. no. 11965092; Thermo Fisher Scientific) supplemented with 10% (v/v) fetal bovine serum (FBS; cat. no. 12483020; Thermo Fisher Scientific), 100 units/ml penicillin-streptomycin (Wisent), and 1X sodium pyruvate (Wisent) at 37 °C in a humidified incubator containing 5% CO$_2$ unless otherwise specified. All cells utilized were periodically tested for mycoplasma contamination using a mycoplasma PCR detection kit (cat. no. G238; Applied Biological Materials) and showed no signs of infection.

For serum starvation, cells were washed three times with PBS after media removal and cultured in DMEM containing 0.5% FBS for the indicated time.

For proteasome inhibition, cells were treated with 20 μM MG132 for 6 h. See Supplementary Data 8 for detailed information of chemicals used for drug treatment.

**Preparation of cell or tissue lysates and immunoblotting.** For tissue lysates, frozen mouse tissues were pulverized in liquid nitrogen followed by homogenization and sonication in buffer K consisting of 20 mM Phosphate Buffer pH 7.0, 15 mM NaCl, 1% NP40, 5 mM EDTA, as well as protease and phosphatase inhibitors (Thermo Fisher Scientific). To prepare liver nuclear extracts, livers were homogenized and subjected to rotation at 4 °C for 1 h. After centrifugation at 94 × g, nuclear pellets were rinsed twice with buffer K followed by sonication in buffer K.

For analyzing whole cell extracts of cultured cells, cells were washed in ice-cold PBS and lysed using buffer K. For cytoplasmic and nuclear fractionation, cells were resuspended with harvest buffer (10 mM Hepes, 50 mM NaCl, 0.5 M Sucrose, 10 mM EDTA, 0.5% Triton X100) and incubated on ice for 5 min. Following centrifugation at 845 × g, the supernatant was removed and centrifuged at 9391 × g at 4 °C to obtain the cytosolic fraction. Nuclear pellets were washed with buffer A (10 mM Hepes, 10 mM KCl, 0.1 mM EDTA, 0.1 mM EGTA) and extracted with high salt buffer (25 mM Hepes, 0.4 M NaCl, 1.5 mM MgCl$_2$, 0.2 mM EDTA, 1% NP40) containing protease and phosphatase inhibitors. Nuclear extracts were collected by centrifugation at 9391 × g at 4 °C for 15 min.

Protein lysates were subjected to determination of concentration using Bio-Rad protein assay dye reagent (cat. no. 5000006). After boiling with SDS sample buffer, denatured proteins were separated by SDS-PAGE, transferred to PVDF membranes (Bio-Rad), blotted according to primary antibody manufacturers' recommendations, and detected using ECL Western blotting detection reagent (Amersham Biosciences). Data were collected using HyBlot CL Autoradiography Film (cat. no. E3018, Harvard Apparatus) together with Epson Perfection V700 Photo (Epson) or using ChemiDoc MP imaging System. The following antibodies were used: ERRα rabbit antibody (1:1000 dilution; cat. no. ab76228; Abcam); Phospho-AKT (Ser473) antibody (1:1000 dilution; cat. no. 4060S; Cell Signaling Technology); AKT antibody (1:1000 dilution; cat. no. 9272S; Cell Signaling Technology); Phospho-GSK-3β (Ser9) antibody (1:1000 dilution; cat. no. 9322S; Cell Signaling Technology); GSK-3β antibody (1:1000 dilution; cat. no. 12456S; Cell Signaling Technology); GSK-3α antibody (1:1000 dilution; cat. no. 4337S; Cell Signaling Technology); Vinculin antibody (1:2000 dilution; cat. no. MAB3574; clone VIIF9; Sigma-Aldrich); Lamin B1 antibody (1:2000 dilution; cat. no. 12586S; Cell Signaling Technology); alpha-Tubulin antibody (1:5000 dilution; cat. no. CLT9002; clone DM1A; Cedarlane); Total OXPHOS Rodent WB Antibody Cocktail (1:5000 dilution; cat. no. ab110413; Abcam); HA antibody (1:500 dilution; cat. no. sc-805; Santa Cruz Biotechnology); beta-Actin antibody (1:2000 dilution; cat. no. ab8226; clone mAbcam 8226; Abcam); CK2α antibody (1:200 dilution; cat. no. sc-373894; clone E-7; Santa Cruz Biotechnology); Flag antibody (1:2000 dilution; cat. no. F1804; clone M2; Sigma-Aldrich); Ubiquitin antibody (1:1000 dilution; cat. no. 3933S; Cell Signaling Technology); FBXW7 antibody (1:1000 dilution; cat. no. ab109617; Abcam); FBXW7 antibody used for studies involving FBXW7 LKO mice (1:1000 dilution; cat.

no. MAB7776; clone # 800201; R&D Systems); V5 antibody (1:2000 dilution; cat. no. ab9116; Abcam); FBXO7 antibody (1:1000 dilution; cat. no. H00025793-M01; clone 4G8; Abnova); FBXO11 antibody (1:500 dilution; cat. no. sc-393229; clone E-9; Santa Cruz Biotechnology); WWP1 antibody (1:1000 dilution; cat. no. LS-C333953; LSBio); Parkin antibody (1:200 dilution; cat. no. sc-32282; clone PRK8; Santa Cruz Biotechnology); Phospho-ERRα (Ser19) antibody[57](1:2000 dilution); Phospho-Akt (Thr308) antibody (1:1000 dilution; cat. no. 9275 S; Cell Signaling Technology); Phospho-Glycogen Synthase (Ser641) antibody (1:2000 dilution; cat. no. 47043T; Cell Signaling Technology); Glycogen Synthase antibody (1:2000 dilution; cat. no. 3886S; Cell Signaling Technology); Rabbit IgG, HRP-Linked Whole Ab (1:5000 dilution; cat. no. CA95017-556L; VWR); Mouse IgG, HRP-Linked Whole Ab (1:5000 dilution; cat. no. CA95017-332L; VWR). For detailed antibody information, see Supplementary Data 8.

SuperSep™ Phos-tag™ (50 μmol/L), 7.5% precast gels (FUJIFILM Wako Pure Chemical Corporation, cat. no. 4548995049858) were used to separate phosphorylated and dephosphorylated ERRα proteins. Gels were agitated gently in a transfer buffer (25 mmol/L Tris, 192 mmol/L Glycine, 10% MeOH) containing 10 mmol/L EDTA for 3*10 min, followed by agitation in a transfer buffer that does not contain EDTA for another 10 minutes before transferring to PVDF membranes. The membranes were blocked with 2% Milk/TBST, reacted with ERRα antibody, and detected by chemiluminescence. Given phosphorylated ERRα protein is bound and trapped by Phos-tag™ during SDS-PAGE, it is separated from dephosphorylated ERRα protein and presents as a slower migrating upper band. Immunoblots were quantified using ImageJ software (with Java 1.8.0_172; National Institutes of Health) if applicable. Images were cropped using Adobe photoshop 2021 and presented in Adobe illustrator 2021.

Uncropped immunoblots presented in main Figures and Supplementary Figures are supplied at the end of the Supplementary Information file with regions used to generate figures bordered with light-dotted lines.

**DNA constructs, vector transduction, and RNA interference**. pCMX_My-c_ERRα_WT and mutants (S19A, S22A) constructs have been previously described[18,58], based on which we further generated ERRα S26A, S26E, S19A + S22A, S19D + S22D, 3SA (S19A + S22A + S26A) mutants via PCR-based site-directed mutagenesis using a Q5® Site-Directed Mutagenesis Kit (cat. no. E0554S; NEB). Vectors were constructed into pLPC_3xFlag or pGEX-5X-3 vectors for transfection of mammalian or bacterial cells, respectively. Short hairpin RNA (shRNA) vectors and pLX317 v5 tagged E3 ligase ORF vectors were from the Mission TRC library (Sigma) and TRC3 ORF collections developed by members of the ORFeome Collaboration (Sigma/TransOMIC), respectively, provided by the Genetic Perturbation Service of Goodman Cancer Research Institute at McGill University.

Calcium phosphate precipitation was used to transiently transfect HEK293T cells. FuGENE HD Transfection Reagent (cat. no. E2311; Promega) or retrovirus were used for vector transduction into HepG2 cells. For retrovirus infection, HEK293T cells were transfected with constructs encoding ERRα WT or 3SA mutant together with helper plasmid for virus production. HepG2 cells were transduced by incubating in retrovirus-containing media supplemented with 8 μg/ml polybrene. Cells were then selected with 1 μg/ml puromycin (Sigma) 30 h post infection and used for experiments. shRNA knockdown was performed according to the protocol as described at http://www.broadinstitute.org/rnai/public/resources/protocols. Briefly, shRNA vectors were transfected into HEK 293T cells together with psPAX2 and pMD2.G vectors for virus production. Experimental cells were infected by incubating in lentivirus-containing media supplemented with 8 μg/ml polybrene and selected with 1 μg/ml puromycin 30 h post infection. siRNA was transfected using the HiPerFect Transfection Reagent (cat. no. 301707; QIAGEN) diluted in Opti-MEM (cat. no. 31985062; Thermo Fisher Scientific).

For detailed information of used plasmids and siRNAs, see Supplementary Data 8.

**Immunoprecipitation (IP)**. For immunoprecipitation studies, cell and tissue lysates were prepared using buffer K supplemented with 0.3% CHAPS. Following determination of protein concentration, equal amounts of protein were incubated with protein A Dynabeads (cat. no. 10008D; Thermo Fisher Scientific) pre-bound with normal rabbit IgG (0.5 μg/IP; cat. no. 10500 C; Invitrogen) or normal mouse IgG (0.5 μg/IP; cat. no. 10400 C; Invitrogen), and the indicated antibodies (0.5 μg/IP). For detailed antibody information, see Supplementary Data 8. After washing with PBS-T (PBS + 0.1% Tween 20), protein complexes were eluted and denatured using SDS sample buffer for immunoblotting analysis as described above. Instead of the conventional HRP-linked rabbit secondary antibody, a special rabbit TrueBlot® IgG (1:4000 dilution; cat. no. 18-8816-33; Rockland) was used to eliminate interference by the heavy and light chains. ERRα immunoprecipitation was performed using a specific anti-hERRα polyclonal antibody previously developed in our laboratory against the N terminus of the protein[58], in a dilution of 1:250, and a mouse ERRα monoclonal antibody (1:1000 dilution; cat. no. NBP2-45523; Clone OTI2C12; Novus Biologicals) was used to detect the immunoprecipitated ERRα protein.

**Protein purification and in vitro kinase assay**. pGEX bacterial expression plasmids encoding GST-ERRα WT or phospho-mutants were transformed into BL21 bacterial cells. Bacteria were incubated at 37 °C until OD 600 nm achieved about 0.6, protein expression was then induced with 0.4 mM IPTG (Wisent) at 30 °C for 4 h. Bacteria pellets were resuspended with STE buffer (10 mM Tris-HCl pH 8.0, 1 mM EDTA, 100 mM NaCl, 5 mM DTT) containing protease and phosphatase inhibitors and incubated on ice with 1 mg/ml lysozyme (Roche) for 30-45 min prior to sonication. Glutathione Sepharose beads (cat. no. 17-0756-01; GE Healthcare) were washed twice with NETN buffer (10 mM Tris-HCl pH 8.0, 1 mM EDTA, 100 mM NaCl, 0.5% NP-40) and incubated with equal amounts of bacterially expressed GST-ERRα recombinant proteins at 4 °C for 2 h with rotation. Beads were washed twice with NETN buffer, once with a high salt (500 mM NaCl) NETN buffer, and one more time with NETN buffer. Following centrifugation, beads were incubated with elution buffer (25 mM glutathione, 50 mM Tris pH 8.8, 200 mM NaCl) and rotated at RT for 30 min before protein elution via centrifugation. GSK3β protein was immunoprecipitated from HEK293T cells transiently overexpressing HA-GSK3β for 48 h. The beads were washed three times with PBST, another three times with kinase buffer (50 mM HEPES pH 7.5, 10 mM MgCl2, 1 mM DTT), and then incubated with the above purified GST-ERRα recombinant proteins in the presence of 5 μCi [γ-$^{32}$P] ATP (cat. no. BLU002-Z250UC; PerkinElmer) and 50 μM ATP (cat. no. PV3227; Thermo Fisher Scientific) in 30 μl of kinase buffer at 30 °C for 1 h. To stop the reactions, tubes were put on ice followed by the addition of SDS sample buffer. Heat-denatured samples were separated on an 8% SDS-PAGE, dried, exposed to Molecular Dynamics Storage Phosphor Screen, and viewed by the Typhoon TRIO Variable Mode Imager (Amersham Biosciences). Gels were then stained with Coomassie brilliant blue to view the input GST-ERRα proteins. Uncropped gels are supplied at the end of the Supplementary Information file with regions used to generate figures bordered with light-dotted lines.

**In vivo adenovirus experiments**. Adenoviruses expressing Cre recombinase (cat. no. 000023A) or GFP (cat. no. 000541A) were purchased, amplified, and purified by Applied Biological Materials Inc. $2 \times 10^9$ plaque-forming unit (pfu) adenovirus was diluted in DMEM and delivered to FBXW7 floxed mice via tail vein injection. Mice were sacrificed 6 days after injection and tissue samples were harvested for analysis.

**Indirect calorimetry**. For simultaneous measurements of whole body metabolic rates, mice fed a ND or a HFD were housed individually in metabolic cages (Sable Systems International, Promethion high-definition behavioral phenotyping system) maintained on a 12 h:12 h light:dark cycle at 22 °C with free access to water and food. VO2, VCO2, RER, locomotor and ambulatory activity, as well as food/water intake were measured after at least 24 h acclimation. For C29 injections, mice were subjected to daily intraperitoneal injection of vehicle or C29 at ZT 10 and placed back in the metabolic cages. CalR[59] (version 1.3) was used to generate hourly plots with gray squares denoting night periods.

**Body composition**. Body composition of live mice were examined using an Echo MRI-100 body composition analyser (EchoMRI LLC).

**Histology**. Liver and eWAT were dissected, fixed for 24 h in 10% formalin (Thermo Fisher Scientific) and then stored in 70% ethanol at 4 °C. Histological procedures were performed at the Histology Core Facility of the Goodman Cancer Research Institute. After embedding in paraffin, tissue fragments were sectioned and stained with hematoxylin and eosin (H&E) or Periodic acid–Schiff (PAS). For Oil Red O (ORO) staining, liver sections were embedded in Optimal cutting temperature (OCT) compound (Thermo Fisher Scientific) followed by frozen sectioning and staining. Stained slides were scanned using Aperio ScanScope XT and viewed by Aperio ImageScope (version 12.4.3.5008; Leica Biosystems). Hepatic accumulation of lipid droplets and PAS positivity were determined using the Aperio digital image analysis algorithms (Aperio Technologies). White adipocyte cell diameters were measured using Aperio ImageScope, four random sections per mouse were quantified. Uncropped histology staining sections are displayed at the end of the Supplementary Information file with regions used to generate figures bordered with light-dotted lines.

**Biochemistry measurements**. Blood samples were collected from the submandibular vein of mice unless otherwise specified. Blood glucose and lactate levels were determined using the OneTouch Ultra®2 glucose meter (LifeScan) and the Lactate Scout meter (Lactate.com), respectively. Blood was then incubated at RT for 30 min before centrifugation at $2348 \times g$ for serum separation. Serum was stored at −80 °C before use. Circulating insulin level was measured using an Ultra Sensitive Mouse Insulin ELISA Kit (cat. no. 90080; Crystal Chem). Serum TG and FFA levels were assessed using a Triglyceride Assay Kit (cat. no. ab65336; Abcam) and a Free Fatty Acid Assay Kit (cat. no. ab65341; Abcam), respectively, as per the manufacturer's recommendations. Serum AST was measured using an Aspartate Aminotransferase (AST/GOT) Activity Assay Kit (cat. no. E-BC-K236-M; Elabscience).

Liver and muscle glycogen contents and PDH enzyme activities were determined using a Glycogen Assay Kit (cat. no. ab65620; Abcam) and a PDH

enzyme activity microplate assay kit (cat. no. Ab109902; Abcam), respectively. Muscle FFA was measured using a Free Fatty Acid Assay Kit (cat. no. ab65341; Abcam).

**Glucose and insulin tolerance tests**. Mice were fasted overnight before the tests with free access to water. The next morning, tolerance to an intraperitoneal injection of glucose or insulin was determined by changes in blood glucose over a period of 120 min following an initial measurement of basal fasting blood glucose levels. ND-fed mice were injected with 2 g/kg glucose for GTTs and 1 unit/kg insulin for ITTs. HFD-fed mice were injected with 1.5 g/kg glucose for GTTs and 2 unit/kg insulin for ITTs. Tail blood samples were measured at 15, 30, 60, 90, and 120 min using the OneTouch Ultra®2 glucose meter.

**RNA extraction and real-time PCR**. Total RNA was extracted from frozen liver and skeletal muscle using a RNeasy Mini Kit (cat. no. 74106; QIAGEN) and a RNeasy Fibrous Tissue Mini Kit (cat. no. 74704; QIAGEN), respectively. RNA was then reverse transcribed using ProtoScript II Reverse Transcriptase (cat. no. M0368X; NEB) according to manufacturer's instructions. Synthesized cDNA was amplified by quantitative real-time PCR (qRT-PCR) with SYBR Green Master Mix (cat. no. 4887352001; Roche) on a LightCycler 480 instrument (Roche). The relative expression levels of selected mouse genes were normalized to the house-keeping gene *Arbp*, *Hprt*, or *B-actin*; human gene expression was normalized to the housekeeping gene *RPLP0*, *TBP*, or *HPRT1*. Specific primer sequences are listed in Supplementary Data 7.

**RNA-sequencing**. Total RNA was extracted as described above. Illumina RNA sequencing and analysis were performed by Novogene Bioinformatics Technology Co.,Ltd. Briefly, RNA was quantified and qualified before generation of the sequencing libraries using NEBNext® UltraTM RNA Library Prep Kit for Illumina® (NEB) following manufacturer's recommendations. After qualification of the library on the Agilent Bioanalyzer 2100 system, samples were clustered on a cBot Cluster Generation System using PE Cluster Kit cBot-HS (Illumina). The library preparations were then sequenced on an Illumina platform and paired-end reads were generated. For data analysis, reference mouse genome mm10 and gene model annotation files were directly downloaded from the genome website browser (http://hgdownload.cse.ucsc.edu/goldenpath/mm10/bigZips/). The STAR (Spliced Transcripts Alignment to a Reference) software[60] (version 2.5) was used to align paired-end clean reads to the reference genome, and to count number reads per gene while mapping. FPKM (Fragments per kilobase of transcript sequence per millions base pairs sequenced) of each gene was calculated based on the length of the gene and read counts mapped to this gene. Differential expression analysis between two groups was performed using the DESeq2 R package:[61] version 1.14.1 for liver RNA-sequencing data of WT/ERRα KO mice treated with saline/insulin, version 2_1.6.3 for liver/muscle RNA-sequencing data of WT/ERRα[3SA] mice. Genes with $P$ value < 0.05 and $|FC| \geq 1.20$ found by DESeq2 were assigned as differentially expressed. DEGs of insulin-stimulated liver/muscle, and insulin-treated HepG2 cells were from the original study for GSE117741[62] and GSE107334[63]. Analyzed RNA-seq data are provided in Supplementary Data 1, 2, 3 and 6. Metascape[64] (version 3.5) was used for functional enrichment of DEGs.

**ChIP-qPCR**. For liver ChIP, two livers from WT or ERRα[3SA] mice were pooled and homogenized in cold PBS containing protease inhibitors using a polytron. Following centrifugation at $845 \times g$ at 4 °C, cell pellets were resuspended in PBS containing 1% formaldehyde and protease inhibitors and subjected to rotation at RT for 12 min to allow DNA-protein crosslinking. Samples were then centrifuged at $845 \times g$ at 4 °C, and cell pellets were washed twice with cold PBS prior to resuspension in cell lysis buffer (5 mM HEPES pH 8, 85 mM KCl, 0.5% NP-40, protease inhibitors) and rotation at 4 °C for 30 min. Nuclear pellets were collected after centrifugation at $845 \times g$ at 4 °C. For muscle ChIP, skeletal muscles from five WT or ERRα[3SA] mice were pooled, homogenized and centrifuged like above. Pellets were resuspended in cell lysis buffer and left to rotate at 4 °C for 30 min. After centrifugation at $845 \times g$ at 4 °C, nuclear pellets were resuspended in PBS containing 1% formaldehyde as well as protease inhibitors and rotated at RT for 12 min. Samples were then centrifuged at $845 \times g$ at 4 °C, and the nuclear pellets were washed twice with cold PBS. Nuclear pellets from liver or muscle were resuspended in nuclei lysis buffer (50 mM Tris-HCl pH 8.1, 10 mM EDTA, 1% SDS, protease inhibitors) and sonicated on ice until the DNA fragments were enriched around 200–500 base pairs.

For ERRα ChIP assays, 10 μl of ERRα antibody (cat. no. ab76228; Abcam) were pre-bound overnight at 4 °C to 60 μl of Dynabeads protein G (cat. no. 10003D; Thermo Fisher Scientific) diluted in 250 μl of blocking buffer (PBS, 0.5% BSA, protease inhibitors). 200 μg chromatin DNA was diluted in 2.5X ChIP dilution buffer (2 mM EDTA pH 8.0, 100 mM NaCl, 20 mM Tris-HCl pH 8.0, 0.5% Triton X-100, protease inhibitors) together with 100 μl of blocking buffer, and added to the antibody-bound beads that were washed three times with blocking buffer and rotated overnight at 4 °C. Beads were washed three times with cold LiCl wash buffer (100 mM Tris-HCl pH 7.5, 500 mM LiCl, 1% NP-40, 1% Na-deoxylcholate), rotating at 4 °C for 3 min per wash. Beads were then transferred to a new tube and

washed two more times with LiCl buffer followed by a brief wash with cold TE buffer (10 mM Tris-HCl pH 7.5, 1 mM EDTA pH 8.0). Beads were decrosslinked in 300 μl of decrosslink buffer (1% SDS, 0.1 M NaHCO$_3$) at 65 °C overnight. Supernatant was collected and incubated with 300 μl TE buffer and 0.2 μg/μl RNase A (QIAGEN) at 37 °C for 1–2 h followed by incubation at 55 °C with 0.2 μg/μl Proteinase K (BioSHOP) for 1–2 h. Chromatin immunoprecipitated DNA was purified using a QIAquick PCR Purification Kit (cat. no. 28106; QIAGEN) and eluted with 35 μl of elution buffer (10 mM Tris-HCl pH 8.0, 0.1 mM EDTA pH 8.0). qPCR was performed as described above to assess the enrichment of ERRα at specific promoters. Enrichment of DNA fragments was normalized against two amplified regions located approximately 4 kb upstream of the ERRα and 49 kb upstream of the PROX1 transcriptional start site. Specific mouse primers used for ChIP-qPCR analysis are shown in Supplementary Data 7.

**Statistics and reproducibility**. GraphPad Prism 9 software was used to generate graphs and for statistical analyses. $F$-test for analysis of covariance (ANCOVA), using body weight as covariate, and one-way analysis of variance (ANOVA) were performed using CalR[59] (version 1.3) for indirect calorimetry studies. D'Agostino-Pearson omnibus normality test was used to determine the normal distribution. The specific statistical tests and definition of error bars are denoted in the corresponding figure legend. For all data, differences were considered significant when $P < 0.05$. "n" values represent individual mice for in vivo experiments or independent biological replicates for cultured cell experiments. Sample size was not pre-specified statistically. For all in vivo experiments, at least three biological replicates were performed and sample sizes underlying each measurement are denoted in the corresponding figure legend. Panels shown without biological replicates are representative of independent experiments. At least three biological replicates were performed for all in vivo experiments. Panels shown without biological replicates are representative of independent experiments. The numbers of experimental repetitions were as follows: Figs. 2a, 2 times; Fig. 2b, c, 3 times; Fig. 2e, i, 2 times; Figs. 3b, 3 times; Figs. 3c, 2 times; Figs. 3f, 3 times; Figs. 3g-i, 2 times; Figs. 3j, 3 times; Figs. 4c, 2 times; Figs. 4d, 3 times; Figs. 4e, 2 times; Figs. 4f, 3 times; Figs. 4g, 2 times; Fig. 4h, i, 3 times; Figs. 4j, 2 times; Fig. 5h, $n = 5$ per group for ND liver H&E and ORO, $n = 7$ per group for HFD liver H&E and ORO, $n = 5$ per group for ND eWAT H&E, $n = 6$ per group for HFD eWAT H&E; Fig. 6e, $n = 4$ for WT, $n = 6$ for 3SA; Figs. 6m, 2 times; Fig. 7n, $n = 4$ for Veh, $n = 5$ for C29; Supplementary Fig. 1a, 2 times; Supplementary Fig. 1c, 3 times; Supplementary Fig. 1e, j, 2 times; Supplementary Fig. 2a, 2 times; Supplementary Fig. 2b, 3 times; Supplementary Fig. 2c, e, 2 times; Supplementary Fig. 3a, b, 2 times; Supplementary Fig. 3d, f, 3 times; Supplementary Fig. 4b–j, 2 times; Supplementary Fig. 5b–d, 2 times; Supplementary Fig. 5e, 3 times; Supplementary Fig. 5h, 2 times; Supplementary Fig. 7f, 2 times.

**Reporting summary**. Further information on research design is available in the Nature Research Reporting Summary linked to this article.

## Data availability

Liver RNA-sequencing data of WT and ERRα KO mice treated with saline or insulin as well as liver/muscle RNA-sequencing data of WT and ERRα[3SA] mice generated in this study have been deposited in Gene Expression Omnibus (GEO) under the accession number GEO: GSE182000. Reference mouse genome mm10 downloaded from the genome website browser (http://hgdownload.cse.ucsc.edu/goldenpath/mm10/bigZips/) was used for data analysis. Analyzed RNA-seq data are provided in Supplementary Data 1, 2 and 6. Mouse ERRα ChIP-seq GEO: GSE43638[16], liver/muscle RNA-sequencing of insulin treatment GEO: GSE117741[62], RNA-sequencing of insulin treatment in HepG2 cells GEO: GSE107334[63], and liver RNA-sequencing of HFD C57BL/6 mice GEO: GSE77625[65] used in this study were downloaded from GEO. Source data are provided with this paper. Any additional information required to reanalyze the data reported in this paper is available from the lead contact upon request. Source data are provided with this paper.

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

## Acknowledgements

We thank members of the Giguère laboratory for their support and helpful discussions; Dr. Mitra Cowen and Ms. Jade Desjardins from the transgenic core at the Goodman Cancer Research Institute (GCRI) for their assistance with the generation of the ERRα mutant mice; the Histology Core Facility at the McGill/GCRI for assistance with processing tissue samples; Ms. Bozena Samborska of Dr. Lawrence Kazak laboratory for assistance with studies involving indirect calorimetry and EchoMRI; and Dr. Arnim Pause for sharing the Alb-Cre mice. This work was supported by a Foundation Grant from the Canadian Institutes of Health Research (FDT-156254) to V.G. H.X. received a Charlotte and Leo Karassik Foundation Oncology PhD Fellowship, a Fonds de Recherche du Québec-Santé (FRQS) Doctoral award, and a Victor K S Lui Fellowship; C.S. is a recipient of a Canderel Fellowship.

## Author contributions

H.X., C.R.D., and V.G. conceived and designed the study; H.X., C.S., and C.R.D. developed the methodology; H.X., C.S., C.O., C.R.D., and M.G. performed the experimental work; H.X. analyzed the data and wrote the original draft of the manuscript, with editing from C.R.D., C.S. and V.G.; V.G. supervised the project and acquired funding.

## Competing interests

The authors declare no competing interests.
