## [Peer Review File · Nature Communications]

REVIEWER COMMENTS

Reviewer #1 (Remarks to the Author):

The manuscript by Xia et al reports data from a substantial body of work that, together, supports a model in which insulin inhibits the kinase GSK3beta, which reduces phosphorylation of the transcriptional regulator ERRalpha. As phosphorylation of ERRalpha promotes its FBXW7-mediated ubiquitination and degradation, this pathway can regulate ERRalpha abundance and control oxidative metabolism.

In general the results are convincing and well described. The data will be of high significance for the regulation of metabolism and for insulin action. There are several points that ought to be clarified and/or further supported prior to publication.

Data for genes used to generate the heat maps in Figs. 1g and 1h should be included in a supplementary spreadsheet. The range for the Z score is given in the figure legend, but it would be helpful to have a scale on the figure itself. It would also be helpful to know the fold change, and not just the Z score. This could be on a spreadsheet.

Spreadsheets should similarly be included for genes identified by ChIP-seq in Extended Data Fig. 1. On page 5, the sentence "Consistently, intersection of transcriptome and cistrome datasets revealed..." is confusing. It appears that the cistrome being referred to is the ChIP-seq data, but this experiment has not yet been introduced.

The model in Fig. 2g is confusing. Phosphorylation activates Akt, but inhibits GSK3b. Perhaps the phosphates can be indicated in different colors, to indicate how they modulate the activity of the target enzyme? The arrow from GSK3b to ERRalpha just indicates "stabilization" but this is actually a lack of phosphorylation and, hence, stabilization. The figure legend could be used to help clarify the model.

In Ext. Data Fig. 3a it is not clear what SE and LE indicate. Presumably, SE is the slower migrating phosphorylated form and LE is the unphosphorylated protein. However this is not clear from the legend, and there is no description of the Phos-tag gel system in the methods section. Also, the two bands are shown on separate panels, so it is not possible to estimate what fraction of the protein is phosphorylated. As well, the authors conclude from this figure that "insulin-stimulated nuclear ERR α protein was unphosphorylated (Extended Data Fig. 3a), indicating that ERR α translocation between the cytoplasm and nucleus is phosphorylation-dependent." This does not follow from the data, which include only nuclear extracts. The conclusion is supported by Ext. Data Fig. 3b, however this relies on transfected cells. Even so, the language could be softened.

None of the immunoblots shown have molecular weight markers indicated, and these should be added to the figures.

Only two GSK3beta consensus sequences are indicated in Fig. 3d, but Fig. 3k diagrams GSK3B phosphorylating three sites. This can be explained if E30 acts as the priming site, as indicated in Fig. 3k,

but then a third consensus should be indicated in Fig. 3d.

As a minor point, the colPs in Ext Data Fig 4e do not necessarily indicate direct interaction, since these were done using transfected cells. Text at top of p. 10 may overstate the interpretation. A similar point is relevant for Fig. 4c; in this case the figure supports the idea that the interaction is physiologic, though not necessarily that it is direct (data in Fig. 4d are more convincing that it is direct, when taken together with previous work). Softening the language would be appropriate.

Fig. 4k and Ext Data Fig. 4j present data from FBXW7 floxed mice treated with adenoCre, however no immunoblots of FBXW7 are shown. This is needed to assess the degree of knockout.

Data in Fig. 5b and 5c extend the model to muscle, as well as liver, but it is not clear whether FBXW7 is present in muscle. Has this been shown in previous work? Or might some other E3 regulate ERR α in muscle?

In Fig. 6, n is given as a range (e.g. n=6-8). This makes interpretation of the statistical tests impossible. Exact n values need to be provided for each group. This is a concern for other figures as well. In some cases, individual data points are shown, but the reader should not have to count these, and in other cases there is only a bar graph with error bars (and so it is especially important to know n for each bar).

Fig. 6 shows that 3SA mice have postprandial insulin resistance, but the impairment in Akt signaling is minimal. Recent data describe the role of TUG cleavage in muscle glucose uptake (PMID: 33686286), and it would be interesting to know if this pathway is impaired. Do the DEGs in 3SA muscle overlap with genes regulated by TUG-C/PPAR γ /PGC-1 α ?

Can an ERR α phosphatase be identified? This is certainly beyond the scope of the present work, but this might be an important regulator. This could be pointed out in the discussion.

Reviewer #2 (Remarks to the Author):

In this manuscript, Xia et al. discovered an insulin-dependent post-translational modification mechanism for ERR α stability and activity that involves ERR α serine phosphorylation by GSK3 β followed by ubiquitination and degradation by FBXW7. This new mechanism is well supported by extensive gain and loss-of-function studies, including in vivo studies using liver GSK3 β KO, liver Fbxw7 KO mice and especially the ERR α 3SA mutant mice. Overall, the findings are significant and the manuscript is well written.

1. Gender is known an important factor in metabolism. For mouse metabolic studies presented in Fig 5e-5l, Fig 6, Extended Fig 5f-fl and Fig 6, male and female mice should be grouped and compared separately. Gender information should also be clearly stated in the figure legend and methods.

2. RNA-seq studies presented in Fig 1g-1h and Extended Fig 1d-1f make an important statement that

much of insulin-mediated gene expression changes is $ERR\alpha$ dependent. However, a $[FC]>1.2$ (20% change) is quite a low bar to define “changed” gene expression. What is the logic of using $[FC]$ of 1.2? If readers are looking for genes $[FC]>2$, what will the Venn diagrams and pathways look like and how much of insulin-mediated gene expression changes are still $ERR\alpha$ dependent? Similar questions are for studies presented in Fig 5d. For Extended Fig 1f, separate pathways for up- and down-regulated genes should be presented (as in Extended Fig 5e) since Fig 1h indicates that genes of the same pathways (carbohydrate/fat metabolism) can be both up- and down-regulated.

3. For experiments shown in Fig 2a-2c and Extended Fig 2a-2c, are there significant amount of insulin in the culture media? As the overall point is $ERR\alpha$ stability in the presence of insulin, these experiments should include conditions with insulin (as in Extended Fig 2d).

4. Additional controls will help the readers. For example, $ERR\alpha$ gene expression (mRNA level) should be included for Fig 1a-1f and Extended Fig 1a-1c to establish whether $ERR\alpha$ transcript is induced in the same experimental conditions.

5. Some experimental details/conclusions shall be clarified. (a) Fig1c labeled “nuclear extracts” on the top – are nuclear extracts used for $ERR\alpha$ only or also for other proteins? (b) Legend of Extended Data Fig 1b states that “Hepatocytes were treated with 0, 100 nM or 10 ug/ml insulin...” – please use consistent units of insulin of either nM or ug/ml in this and other figures and throughout the manuscript. (c) The authors claim that “insulin-stimulated nuclear $ERR\alpha$ protein was unphosphorylated (Extended Data Fig. 3a)”, but the short exposure (SE, supposedly more accurate) phospho- $ERR\alpha$ level seems clearly increased. (d) In Fig 3g, why do 3SA mutations affect GSK3 β S9 phosphorylation?

Liming Pei

Reviewer #3 (Remarks to the Author):

Xia and colleagues showed insulin controls the activity of $ERR\alpha$ via GSK3b/FBXW7 axis in vitro. They demonstrated $ERR\alpha$ is phosphorylated by GSK3b, downstream target of PI3K-AKT pathway, and ubiquitylated by FBXW7 leading to proteasome dependent degradation. In the latter part of the manuscript, they generated mice with $ERR\alpha$ 3SA, and found that $ERR\alpha$ 3SA mice develop insulin resistance, which showed the biological significance of $ERR\alpha$ in vivo.

Overall, the authors have done an excellent analysis, and revealed that $ERR\alpha$ is one of key molecules that control energy homeostasis.

Specific points:

1. In Extended Fig 1h, 7079 targets of $ERR\alpha$ ChIP-seq are shown, and it seems that these targets are from 20 kb upstream and downstream of genes. Should the targets from 20 kb downstream of genes be included in this analysis because $ERR\alpha$ is a transcriptional factor? Do the 7079 targets include multiple sites from one gene?

2. Alb-Cre and Gsk3b^{f/f} mice are both controls, but $ERR\alpha$ expressions are so different between them.

This figure is confusing (Figure 2d). Also, it is better to show the data from Alb-Cre/Gsk3b+/f.

3. The accumulation of ERR α protein is nicely shown in Figure 2e and 2f, however, mRNA levels should be examined to indicate that these are caused by post-translational modification, not by transcriptional level.

4. It is difficult to see the result in Figure 3i, upper right panel, because upper part of the picture is cut off.

5. Is there any change in endogenous ERR α protein level after MG132 treatment in HepG2 cell (Figure 3)?

6. The authors showed FBXW7 is a ubiquitin ligase of ERR α in addition to Stub1 and Parkin. Which ligase functions mainly as an ubiquitin ligase of ERR α in vivo (Figure 4)?

7. There is no FBXW7 immunoblot in Figure 4c.

8. In case of analysis of conditional knockout mice, controls should be Alb-Cre or Alb-Cre/Fbxw7+/f (Fig 4k).

9. There are three isoforms in FBXW7. FBXW7 α is ubiquitously expressed, and FBXW7 γ is expressed in muscle. Which isoform is an ubiquitin ligase of ERR α in muscle (Figure 5b)?

10. The time points with significant difference are varied in the night time, although C29 was injected at the same time (Figure 7k). Is there any reason for this?

REVIEWER COMMENTS

Reviewer #1 (Remarks to the Author):

The manuscript by Xia et al reports data from a substantial body of work that, together, supports a model in which insulin inhibits the kinase GSK3beta, which reduces phosphorylation of the transcriptional regulator ERRalpha. As phosphorylation of ERRalpha promotes its FBXW7-mediated ubiquitination and degradation, this pathway can regulate ERRalpha abundance and control oxidative metabolism.

In general the results are convincing and well described. The data will be of high significance for the regulation of metabolism and for insulin action. There are several points that ought to be clarified and/or further supported prior to publication.

Data for genes used to generate the heat maps in Figs. 1g and 1h should be included in a supplementary spreadsheet. The range for the Z score is given in the figure legend, but it would be helpful to have a scale on the figure itself. It would also be helpful to know the fold change, and not just the Z score. This could be on a spreadsheet.

Spreadsheets should similarly be included for genes identified by ChIP-seq in Extended Data Fig. 1. On page 5, the sentence “Consistently, intersection of transcriptome and cistrome datasets revealed...” is confusing. It appears that the cistrome being referred to is the ChIP-seq data, but this experiment has not yet been introduced.

The model in Fig. 2g is confusing. Phosphorylation activates Akt, but inhibits GSK3b. Perhaps the phosphates can be indicated in different colors, to indicate how they modulate the activity of the target enzyme? The arrow from GSK3b to ERRalpha just indicates “stabilization” but this is actually a lack of phosphorylation and, hence, stabilization. The figure legend could be used to help clarify the model.

In Ext. Data Fig. 3a it is not clear what SE and LE indicate. Presumably, SE is the slower migrating phosphorylated form and LE is the unphosphorylated protein. However this is not clear from the legend, and there is no description of the Phos-tag gel system in the methods section. Also, the two bands are shown on separate panels, so it is not possible to estimate what fraction of the protein is phosphorylated. As well, the authors conclude from this figure that “insulin-stimulated nuclear ERR α protein was unphosphorylated (Extended Data Fig. 3a), indicating that ERR α translocation between the cytoplasm and nucleus is phosphorylation-dependent.” This does not follow from the data, which include only nuclear extracts. The conclusion is supported by Ext. Data Fig. 3b, however this relies on transfected cells. Even so, the language could be softened.

None of the immunoblots shown have molecular weight markers indicated, and these should be added to the figures.

Only two GSK3beta consensus sequences are indicated in Fig. 3d, but Fig. 3k diagrams GSK3B phosphorylating three sites. This can be explained if E30 acts as the priming

site, as indicated in Fig. 3k, but then a third consensus should be indicated in Fig. 3d.

As a minor point, the coIPs in Ext Data Fig 4e do not necessarily indicate direct interaction, since these were done using transfected cells. Text at top of p. 10 may overstate the interpretation. A similar point is relevant for Fig. 4c; in this case the figure supports the idea that the interaction is physiologic, though not necessarily that it is direct (data in Fig. 4d are more convincing that it is direct, when taken together with previous work). Softening the language would be appropriate.

Fig. 4k and Ext Data Fig. 4j present data from FBXW7 floxed mice treated with adenoCre, however no immunoblots of FBXW7 are shown. This is needed to assess the degree of knockout.

Data in Fig. 5b and 5c extend the model to muscle, as well as liver, but it is not clear whether FBXW7 is present in muscle. Has this been shown in previous work? Or might some other E3 regulate ERRalpha in muscle?

In Fig. 6, n is given as a range (e.g. n=6-8). This makes interpretation of the statistical tests impossible. Exact n values need to be provided for each group. This is a concern for other figures as well. In some cases, individual data points are shown, but the reader should not have to count these, and in other cases there is only a bar graph with error bars (and so it is especially important to know n for each bar).

Fig. 6 shows that 3SA mice have postprandial insulin resistance, but the impairment in Akt signaling is minimal. Recent data describe the role of TUG cleavage in muscle glucose uptake (PMID: 33686286), and it would be interesting to know if this pathway is impaired. Do the DEGs in 3SA muscle overlap with genes regulated by TUG-C/PPARgamma/PGC-1alpha?

Can a ERRalpha phosphatase be identified? This is certainly beyond the scope of the present work, but this might be an important regulator. This could be pointed out in the discussion.

Reviewer #2 (Remarks to the Author):

In this manuscript, Xia et al. discovered an insulin-dependent post-translational modification mechanism for ERR α stability and activity that involves ERR α serine phosphorylation by GSK3 β followed by ubiquitination and degradation by FBXW7. This new mechanism is well supported by extensive gain and loss-of-function studies, including in vivo studies using liver GSK3 β KO, liver Fbxw7 KO mice and especially the ERR α 3SA mutant mice. Overall, the findings are significant and the manuscript is well written.

1. Gender is known an important factor in metabolism. For mouse metabolic studies presented in Fig 5e-5l, Fig 6, Extended Fig 5f-fl and Fig 6, male and female mice should be grouped and compared separately. Gender information should also be clearly stated in the figure legend and methods.
2. RNA-seq studies presented in Fig 1g-1h and Extended Fig 1d-1f make an important statement that much of insulin-mediated gene expression changes is ERR α dependent. However, a [FC]>1.2 (20% change) is quite a low bar to define “changed” gene expression. What is the logic of using [FC] of 1.2? If readers are looking for genes [FC]>2, what will the Venn diagrams and pathways look like and how much of insulin-mediated gene expression changes are still ERR α dependent? Similar questions are for studies presented in Fig 5d. For Extended Fig 1f, separate pathways for up- and down-regulated genes should be presented (as in Extended Fig 5e) since Fig 1h indicates that genes of the same pathways (carbohydrate/fat metabolism) can be both up- and down-regulated.
3. For experiments shown in Fig 2a-2c and Extended Fig 2a-2c, are there significant amount of insulin in the culture media? As the overall point is ERR α stability in the presence of insulin, these experiments should include conditions with insulin (as in Extended Fig 2d).
4. Additional controls will help the readers. For example, ERR α gene expression (mRNA level) should be included for Fig 1a-1f and Extended Fig 1a-1c to establish whether ERR α transcript is induced in the same experimental conditions.
5. Some experimental details/conclusions shall be clarified. (a) Fig1c labeled “nuclear extracts” on the top – are nuclear extracts used for ERR α only or also for other proteins? (b) Legend of Extended Data Fig 1b states that “Hepatocytes were treated with 0, 100 nM or 10 ug/ml insulin...” – please use consistent units of insulin of either nM or ug/ml in this and other figures and throughout the manuscript. (c) The authors claim that “insulin-stimulated nuclear ERR α protein was unphosphorylated (Extended Data Fig. 3a)”, but the short exposure (SE, supposedly more accurate) phospho-ERR α level seems clearly increased. (d) In Fig 3g, why do 3SA mutations affect GSK3 β S9 phosphorylation?

Liming Pei

Reviewer #3 (Remarks to the Author):

Xia and colleagues showed insulin controls the activity of $ERR\alpha$ via GSK3b/FBXW7 axis in vitro. They demonstrated $ERR\alpha$ is phosphorylated by GSK3b, downstream target of PI3K-AKT pathway, and ubiquitylated by FBXW7 leading to proteasome dependent degradation. In the latter part of the manuscript, they generated mice with $ERR\alpha3SA$, and found that $ERR\alpha3SA$ mice develop insulin resistance, which showed the biological significance of $ERR\alpha$ in vivo.

Overall, the authors have done an excellent analysis, and revealed that $ERR\alpha$ is one of key molecules that control energy homeostasis.

Specific points:

1. In Extended Fig 1h, 7079 targets of $ERR\alpha$ ChIP-seq are shown, and it seems that these targets are from 20 kb upstream and downstream of genes. Should the targets from 20 kb downstream of genes be included in this analysis because $ERR\alpha$ is a transcriptional factor? Do the 7079 targets include multiple sites from one gene?
2. Alb-Cre and Gsk3b^{f/f} mice are both controls, but $ERR\alpha$ expressions are so different between them. This figure is confusing (Figure 2d). Also, it is better to show the data from Alb-Cre/Gsk3b^{+/f}.
3. The accumulation of $ERR\alpha$ protein is nicely shown in Figure 2e and 2f, however, mRNA levels should be examined to indicate that these are caused by post-translational modification, not by transcriptional level.
4. It is difficult to see the result in Figure 3i, upper right panel, because upper part of the picture is cut off.
5. Is there any change in endogenous $ERR\alpha$ protein level after MG132 treatment in HepG2 cell (Figure 3)?
6. The authors showed FBXW7 is a ubiquitin ligase of $ERR\alpha$ in addition to Stub1 and Parkin. Which ligase functions mainly as an ubiquitin ligase of $ERR\alpha$ in vivo (Figure 4)?
7. There is no FBXW7 immunoblot in Figure 4c.
8. In case of analysis of conditional knockout mice, controls should be Alb-Cre or Alb-Cre/Fbxw7^{+/f} (Fig 4k).
9. There are three isoforms in FBXW7. FBXW7 α is ubiquitously expressed, and FBXW7 γ is expressed in muscle. Which isoform is an ubiquitin ligase of $ERR\alpha$ in muscle (Figure 5b)?
10. The time points with significant difference are varied in the night time, although C29 was injected at the same time (Figure 7k). Is there any reason for this?

Response to NCOMMS-21-40051-T comments

Reviewer #1 (Remarks to the Author):

The manuscript by Xia et al reports data from a substantial body of work that, together, supports a model in which insulin inhibits the kinase GSK3beta, which reduces phosphorylation of the transcriptional regulator ERRalpha. As phosphorylation of ERRalpha promotes its FBXW7-mediated ubiquitination and degradation, this pathway can regulate ERRalpha abundance and control oxidative metabolism.

In general the results are convincing and well described. The data will be of high significance for the regulation of metabolism and for insulin action. There are several points that ought to be clarified and/or further supported prior to publication.

We appreciate that the reviewer recognizes the significance of our study and provides us with helpful comments. We have carefully addressed all points raised in our revised manuscript. Please see our detailed response below.

1. Data for genes used to generate the heat maps in Figs. 1g and 1h should be included in a supplementary spreadsheet. The range for the Z score is given in the figure legend, but it would be helpful to have a scale on the figure itself. It would also be helpful to know the fold change, and not just the Z score. This could be on a spreadsheet.

We thank the reviewer for pointing this out. We presented the specific Z score and fold change of genes used to generate original Figs. 1g and 1h (**Now Figs. 1m and 1n**) in **Supplementary Table 1** (Insulin-regulated hepatic DEGs in WT and ERR α KO mice) and **Supplementary Table 2** (Insulin-regulated hepatic gene sets in WT and ERR α KO mice), respectively. We also added the gene lists of RNA-seq performed in **Fig. 5d** as **Supplementary Table 6** (ERR α 3SA liver and muscle Insulin-sensitive DEGs).

Further, we specified the Z score scale for all the presented heatmaps including original Figs. 1g and 1h (**Now Figs. 1m and 1n**), as well as Extended Data Fig. 6m (**Now Supplementary Fig. 6m**).

2. Spreadsheets should similarly be included for genes identified by ChIP-seq in Extended Data Fig. 1. On page 5, the sentence “Consistently, intersection of transcriptome and cistrome datasets revealed....” is confusing. It appears that the cistrome being referred to is the ChIP-seq data, but this experiment has not yet been introduced.

Regarding the cistrome datasets, we previously stated in the Supplementary legend as ERR α ChIP-seq data (GSE43638), which was done in our published study (PMID: 23562079). We apologize for not describing it clearly in the main text and have clarified this point in the revised manuscript.

We also presented the genes identified by ChIP-seq studies in original Extended Data Fig. 1h, i (**Now Supplementary Fig. 1k, l**) in **Supplementary Table 3** (ERR α -bound insulin-sensitive DEGs in HepG2 cells) and **Supplementary Table 4** (ERR α , SREBP1, Foxo1 liver ChIP-seq targets), respectively.

3. The model in Fig. 2g is confusing. Phosphorylation activates Akt, but inhibits GSK3 β . Perhaps the phosphates can be indicated in different colors, to indicate how they modulate the activity of the target enzyme? The arrow from GSK3 β to ERR α just indicates “stabilization” but this is actually a lack of phosphorylation and, hence, stabilization. The figure legend could be used to help clarify the model.

We fully agree with the reviewer on this point and have revised the model in original Fig. 2g (**Now Fig. 2j**) as suggested. We’ve colored the activating phosphate for AKT in green and the inhibiting phosphate for GSK3 β in black. We’ve also added an “Activation” beside the glowing p-AKT and added an “Inactivation” beside the p-GSK3 β . Please see below.

Fig. 2j

We have revised the legend as “j Proposed model of insulin-mediated stabilization of ERR α protein through the PI3K/AKT/ GSK3 β pathway.” We did not state ERR α phosphorylation in this legend, because we only proved that GSK3 β kinase activity is required for insulin-mediated stabilization of ERR α protein in Fig 2, and the true effects of GSK3 β on ERR α phosphorylation status were not shown until Fig 3. However, the revised model presents this possibility using greyed phosphorylation sites.

4. In Ext. Data Fig. 3a it is not clear what SE and LE indicate. Presumably, SE is the slower migrating phosphorylated form and LE is the unphosphorylated protein. However this is not clear from the legend, and there is no description of the Phos-tag gel system in the methods section. Also, the two bands are shown on separate panels, so it is not possible to estimate what fraction of the protein is phosphorylated. As well, the authors conclude from this figure that “insulin-stimulated nuclear ERR α protein was unphosphorylated (Extended Data Fig. 3a), indicating that ERR α translocation between the cytoplasm and nucleus is phosphorylation-dependent.” This does not follow from the data, which include only nuclear extracts. The conclusion is supported by Ext. Data Fig. 3b, however this relies on transfected cells. Even so, the language could be softened.

We apologize for the unexplained abbreviations (SE, LE) and confusing statement. In the original Extended Data Fig.3a (see below), the two bands revealed in the phos-tag gel are the same membrane with a short exposure (SE; middle panel) and longer exposure (LE, lower panel), we could not observe a band shift with the nuclear extract even using longer exposure, thus we previously stated in the main text that “insulin-stimulated nuclear ERR α protein was unphosphorylated”.

We optimized this experiment by adding whole cell extract (WCE) and cytoplasmic fraction as controls and properly labeled the bands (**New Supplementary Fig. 3a**). As shown below, in comparison with whole-cell extract, phos-tag gel examination of the nuclear extract from HepG2 cells could not reveal an upper phosphorylated ERR α band shift, indicating that insulin-induced nuclear ERR α protein was dephosphorylated. This was followed by introducing mutations to the insulin-sensitive ERR α residues which showed the nuclear export of ERR α exclusively with the phospho-mimicking ERR α mutant, implying that ERR α translocation between the cytoplasm and nucleus is mediated by phosphorylation (Supplementary Fig. 3b). We have modified the main text and softened the language as suggested.

Original Extended Data Fig. 3a

Revised Supplementary Fig. 3a

Further, methods and mechanism for the phos-tag gel were added in the “Preparation of cell or tissue lysates and immunoblotting” section of Method: SuperSep™ Phos-tag™ (50μmol/L), 7.5% precast gels (FUJIFILM Wako Pure Chemical Corporation, Cat. No. 4548995049858) were used to separate phosphorylated and dephosphorylated ERRα proteins. Gels were agitated gently in a transfer buffer (25 mmol/L Tris, 192 mmol/L Glycine, 10% MeOH) containing 10 mmol/L EDTA for 3*10 minutes, followed by agitation in a transfer buffer that does not contain EDTA for another 10 minutes before transferring to PVDF membranes. The membranes were blocked with 2% Milk/TBST, reacted with ERRα antibody, and detected by chemiluminescence. Given phosphorylated ERRα protein is bound and trapped by Phos-tag™ during SDS-PAGE, it is separated from dephosphorylated ERRα protein and presents as a slower migrating upper band.

Correspondingly, we pinpointed the two ERRα bands revealed by phos-tag gel as phosphorylated (slow-migrating upper band) and dephosphorylated (lower band) in **Fig 3b, c**. See below.

Fig. 3b

Fig. 3c

5. None of the immunoblots shown have molecular weight markers indicated, and these should be added to the figures.

We thank the reviewer for pointing this out. We have now added the molecular weight markers for all the presented blots.

6. Only two GSK3beta consensus sequences are indicated in Fig. 3d, but Fig. 3k diagrams GSK3B phosphorylating three sites. This can be explained if E30 acts as the priming site, as indicated in Fig. 3k, but then a third consensus should be indicated in Fig. 3d.

We agree with the reviewer and have now included the third putative GSK3β consensus phosphorylation motif in Fig. 3d, see below. We also revised it correspondingly in the main text.

7. As a minor point, the coIPs in Ext Data Fig 4e do not necessarily indicate direct interaction, since these were done using transfected cells. Text at top of p. 10 may overstate the interpretation. A similar point is relevant for Fig. 4c; in this case the figure supports the idea that the interaction is physiologic, though not necessarily that it is direct (data in Fig. 4d are more convincing that it is direct, when taken together with previous work). Softening the language would be appropriate.

As suggested, we revised the description of co-IPs in Ext Data Fig 4e as “Co-IP experiments confirmed the interaction between the 6 E3 candidates and ERRα (Supplementary Fig. 4e).” Similarly, we revised the description for Fig. 4c as “Using hepatocytes, we confirmed that endogenous FBXW7 and ERRα physiologically interact with each other (Fig. 4c).”

8. Fig. 4k and Ext Data Fig. 4j present data from FBXW7 floxed mice treated with adenoCre, however no immunoblots of FBXW7 are shown. This is needed to assess the degree of knockout.

We thank the reviewer for raising this valid point, we have now confirmed this with immunoblotting as shown in **Fig. 4k** and original Ext Data Fig. 4j (**Now Supplementary Fig. 4k**). We also validated the knockdown via RT-PCR using a primer specifically targeting the floxed FBXW7 exon (**New Fig. 4l**). See below.

Fig. 4k**Supplementary Fig. 4k****Fig. 4l**
9. Data in Fig. 5b and 5c extend the model to muscle, as well as liver, but it is not clear whether FBXW7 is present in muscle. Has this been shown in previous work? Or might some other E3 regulate ERRα in muscle?

FBXW7 exists as three protein isoforms (α, β, and γ) that differ in subcellular locations: FBXW7α in the nucleoplasm, FBXW7β in the cytoplasm, and FBXW7γ in the nucleolus. FBXW7α is functionally the most dominant isoform, which is ubiquitously expressed and carries out the most known FBXW7 functions (reviewed in PMID: 25314076).

We first confirmed that ERRα protein was stabilized in C2C12 myoblast cells upon proteasome inhibition without altering its mRNA level (**New Supplementary Fig. 5e, f**). We then assessed and observed comparable expression levels of FBXW7α mRNA in liver and muscle (**New Supplementary Fig. 5g**). See below.

Supplementary Fig. 5
Also, other potential ERRα E3 ligases we identified in Fig. 4 (FBXO7, FBXO11, WWP1), as well as previously published E3 ligases targeting ERRα (STUB1, PARKIN), regulate ERRα stability independent of its phosphorylation at S19, 22, 26, as their interactions with ERRα were not affected by the phospho-defective 3SA mutations (**New Supplementary Fig. 4h**).

Supplementary Fig. 4h

Thus, we speculate the accumulated ERRα protein in the muscle of ERRα^{3SA} mice is caused by the conservation of the insulin-GSK3β-FBXW7-ERRα axis in muscle, which remain to be further validated.

10. In Fig. 6, n is given as a range (e.g. n=6-8). This makes interpretation of the statistical tests impossible. Exact n values need to be provided for each group. This is a concern for other figures as well. In some cases, individual data points are shown, but the reader should not have to count these, and in other cases there is only a bar graph with error bars (and so it is especially important to know n for each bar).

We thank the reviewer for raising these valid points. To address these concerns, we have specified the exact sample sizes in related figures, we've also revised the bar graphs to show Individual data points.

11. Fig. 6 shows that 3SA mice have postprandial insulin resistance, but the impairment in Akt signaling is minimal. Recent data describe the role of TUG cleavage in muscle glucose uptake (PMID: 33686286), and it would be interesting to know if this pathway is impaired. Do the DEGs in 3SA muscle overlap with genes regulated by TUG-C/PPARγ/PGC-1α?

Indeed, the attenuation of refeeding-induced AKT phosphorylation is minimal in ERRα^{3SA} muscle, but the effects of insulin on GSK3β and GS phosphorylation were mostly blunted in ERRα^{3SA} muscle, accounting for its parallel defect in muscle postprandial glycogenesis (Fig. 6c, d).

We fully agree with the reviewer that other PI3K-AKT independent glucose uptake mechanisms might also contributed to the postprandial insulin resistance in ERRα^{3SA} muscle.

For example, insulin-stimulated TUG cleavage facilitates GLUT4 translocation to the cell surface and increase muscle glucose uptake, which is independent of PI3K activity. Here, we examined TUG cleavage by immunoblots using a Tug Antibody (Cell signaling #2049) and observed impaired refeeding-induced TUG cleavage in ERRα^{3SA} muscle (see below **Fig. for Reviewer 1**).

Fig. for Reviewer 1

Given TUG C-terminal cleavage product enters the nucleus, binds PPAR γ /PGC-1 α and regulates gene expression to promote lipid oxidation and thermogenesis, we further overlapped the DEGs identified in ERR α ^{3SA} muscle with DEGs identified in muscle of transgenic mice with constitutive TUG cleavage in muscle (GSE134846). By using the threshold $p < 0.05$ and $|\text{fold change}| \geq 1.20$, we identified 119 common genes (13.5% of total ERR α ^{3SA} muscle DEGs), which are involved in fatty acid metabolism (*Pdk4*, *Ppara*, *Slc25a1*, *Acly*, *Fasn*, *Scd2*, *Angptl4*, *Mfsd2a*), insulin signaling (*Shc2*, *Insig1*), and mitochondrial DNA synthesis (*Cmpk2*).

We did not include these findings in the main text considering the word limitations, but these data indicate that ERR α is also involved in other pathways of muscle glucose uptake, which will be interesting to explore. We have modified our discussion to acknowledge other potential glucose uptake mechanisms: “It will be interesting to explore whether ERR α is responsible for insulin-stimulated glucose disposal in skeletal muscle that are independent of the PI3K-AKT signaling pathway, as attenuation of refeeding-induced AKT phosphorylation is minimal in ERR α ^{3SA} muscle, and multiple studies revealed additional pathways perturbing muscle glucose tolerance and insulin sensitivity in addition to attenuated AKT signaling.”

12. Can a ERRalpha phosphatase be identified? This is certainly beyond the scope of the present work, but this might be an important regulator. This could be pointed out in the discussion.

We agree with the reviewer that this is an important point. The phosphatase(s) involved in ERR α dephosphorylation remains elusive and is a derived project of this manuscript. We speculate that phosphatase calcineurin and PP2A might be involved in the regulation of ERR α phosphorylation and subcellular localization and are currently validating the hypothesis using pharmacological inhibitors. Especially, since PP2A is known to dephosphorylate GSK3 β (PMID: 26484916), it might be of interest to investigate whether it is also involved in the insulin/GSK3 β /FBXW7/ERR α axis.

As suggested, we have reflected this point in the discussion “Alternatively, the phosphatase(s) involved in ERR α dephosphorylation remains elusive. Identification of ERR α phosphatase is essential and left to be explored. Especially since PP2A is known

to dephosphorylating GSK3 β , it might be of interest to investigate whether it is involved in the insulin/GSK3 β /FBXW7/ERR α axis”

Reviewer #2 (Remarks to the Author):

In this manuscript, Xia et al. discovered an insulin-dependent post-translational modification mechanism for ERR α stability and activity that involves ERR α serine phosphorylation by GSK3 β followed by ubiquitination and degradation by FBXW7. This new mechanism is well supported by extensive gain and loss-of-function studies, including in vivo studies using liver GSK3 β KO, liver Fbxw7 KO mice and especially the ERR α 3SA mutant mice. Overall, the findings are significant and the manuscript is well written.

We thank the reviewer for carefully reviewing our manuscript and for the helpful suggestions. We have addressed the points raised. Please see our detailed response below.

1. Gender is known an important factor in metabolism. For mouse metabolic studies presented in Fig 5e-5l, Fig 6, Extended Fig 5f-fl and Fig 6, male and female mice should be grouped and compared separately. Gender information should also be clearly stated in the figure legend and methods.

We fully agree with the reviewer that gender is an important factor in metabolism. In this study all the mouse data were derived from 2- to 3-month-old male littermates except for Supplementary Fig. 6, where we confirmed that the hyperglycemia phenotype is conserved in female mice.

We stated it in the “Mice” section of “Method”: “Unless otherwise specified, all experiments used age-matched male littermates (2- to 3-month-old).”

We also specified the gender of mice used for Supplementary Fig. 6 in the legend: “Fed blood glucose concentrations of 3-month-old female ERR α ^{3SA} and WT littermates on a ND (WT, n = 7; 3SA, n = 6).”

2. RNA-seq studies presented in Fig 1g-1h and Extended Fig 1d-1f make an important statement that much of insulin-mediated gene expression changes is ERR α dependent. However, a [FC]>1.2 (20% change) is quite a low bar to define “changed” gene expression. What is the logic of using [FC] of 1.2? If readers are looking for genes [FC]>2, what will the Venn diagrams and pathways look like and how much of insulin-mediated gene expression changes are still ERR α dependent? Similar questions are for studies presented in Fig 5d. For Extended Fig 1f, separate pathways for up- and down-regulated genes should be presented (as in Extended Fig 5e) since Fig 1h indicates that genes of the same pathways (carbohydrate/fat metabolism) can be both up- and down-regulated.

We used 20% change to identify insulin-sensitive DEGs mainly because insulin had moderate effects on a large number of genes, consistent with the widely-accepted notion that transcriptional changes of individual metabolic genes are small. This has been recently revealed in a paper published in Cell (PMID: 30955890): “insulin had moderate effects on a large number of genes (Figure 4I), consistent with previous reports (Cai et al., 2017), and with insulin’s role as a homeostatic factor broadly modulating metabolism and cell growth.”

As shown below **Fig. for Reviewer 2**, transcriptional changes of most of the insulin-stimulated genes are small.

Fig. for Reviewer 2

Fig. 4I of PMID: 30955890

Also, we can significantly validate a 20% change in gene expression via RT-PCR and a collection of 20% changes in multiple genes enriched in the same metabolic pathway is highly significant. Thus, we used $[FC] \geq 1.2$ for the analysis of RNA-seq data in Fig. 1 and Fig. 5.

In our study presented in original Fig 1g (**Now Fig. 1m**), when we increased the $[FC]$ from 1.2 to 2.0, the insulin-regulated genes reduced from 2455 to 170, supporting the importance of using a less stringent cut-off for insulin-sensitive metabolic genes. Among these 170 genes, 125 are dependent on $ERR\alpha$, thus the percentage of $ERR\alpha$ -dependent insulin-regulated genes increased from 67.4% to 73.5%, indicating the importance of $ERR\alpha$ in insulin-mediated gene expression. See below **Fig. for Reviewer 3** for the Venn’s diagram and heatmap using $[FC] \geq 2.0$ as the cut-off.

Fig. for Reviewer 3

Similarly, for Fig. 5d, if we use $[FC] \geq 2.0$ instead of 1.2, we'll lose 75.4% $ERR\alpha^{3SA}$ liver DEGs and 62.9% $ERR\alpha^{3SA}$ muscle DEGs (See below **Fig. for Reviewer 4**), making it hard to reveal the enriched pathway and the metabolic roles of $ERR\alpha$.

Fig. for Reviewer 4

We further separated the enriched pathway for up- and down-regulated genes for original Extended Fig 1f (**Now Supplementary Fig. 1i**), see below.

Supplementary Fig. 1i

Term	Count	P (-log ₁₀)
Oxidative phosphorylation	29	20.45
Non-alcoholic fatty liver disease (NAFLD)	23	12.85
Mitochondrion organization	34	8.66
Mitochondrial biogenesis	8	7.07
Amino Acid metabolism	13	6.90
Carbohydrate metabolic process	27	3.94
Regulation of cell growth	22	3.69
Fatty acid metabolic process	19	2.83
Citrate cycle (TCA cycle)	5	2.67
Response to insulin	12	2.23

Term (Up DEGs)	Count	P (-log ₁₀)
TCA cycle and respiratory electron transport	25	19.96
Non-alcoholic fatty liver disease	22	17.15
Mitochondrion organization	31	14.98
Mitochondrial biogenesis	8	9.15
Triglyceride biosynthetic process	4	4.24

Term (Down DEGs)	Count	P (-log ₁₀)
Cell junction organization	20	6.48
Liver development	7	4.01
Cellular component morphogenesis	16	3.08
Carbohydrate homeostasis	7	2.13
Protein polyubiquitination	7	2.03

3. For experiments shown in Fig 2a-2c and Extended Fig 2a-2c, are there significant amount of insulin in the culture media? As the overall point is $ERR\alpha$ stability in the presence of insulin, these experiments should include conditions with insulin (as in Extended Fig 2d).

For Fig 2a-2c and Extended Fig 2a-2c, cells were cultured in optimal medium supplemented with 10% Fetal bovine serum, composing growth factors and hormones to sustain proliferation and maintain normal cell metabolism. Thus, this mimics the physiological fed state, neither starved or insulin stimulated. As we can see in Extended Fig 2c, the insulin signaling pathway is active in this condition, thus manipulating kinase activity itself would affect ERR α stability if the kinase is involved in the regulation of ERR α stability downstream the insulin signaling. Thus, we performed the initial kinase screen in the physiological state without manipulating the presence or absence of insulin.

After identifying that GSK3 β inhibition best replicated the effect of insulin on ERR α *in vitro* and *in vivo*, we further validate whether ERR α could still be stabilized by insulin in the absence of GSK3 β (Original Fig. 2e, f ; **Now Fig. 2e, g**) or upon blocking the insulin signaling transduction to GSK3 β by AKT inhibitors (Original Extended Fig 2d; **Now Supplementary Fig. 2e**).

We fully agree that it will be more convincing to add the with / without insulin conditions for Fig. 2c. To rule out the effect of insulin on endogenous GSK3 β , we performed this experiment with cells stably expressing either control or shRNA targeting the 3'UTR of GSK3 β . As shown below (**New Fig. 2i**), re-expression of the constitutively active GSK3 β S9A but not the kinase defective K85A mutant degraded the accumulated ERR α protein in the GSK3 β knock-down cells (compare lanes 9, 11 to lanes 1, 7), and insulin had no further effects on ERR α protein stability once GSK3 β could not be phosphorylated by insulin or lose its kinase activity (compare lanes 9, 10, 11, 12 to lanes 7, 8). Together, our results clearly demonstrate that GSK3 β kinase activity is required for insulin control of ERR α protein stability.

4. Additional controls will help the readers. For example, ERR α gene expression (mRNA level) should be included for Fig 1a-1f and Extended Fig 1a-1c to establish whether ERR α transcript is induced in the same experimental conditions.

We agree with the reviewer that ERR α mRNA expression data are important controls and have conducted the examinations for the original Fig 1a-1f and Extended Fig 1a-1c, which showed that these conditions stabilize ERR α proteins without affecting its mRNA levels. Please see the **New Fig. 1b, d, f, h, j, l** and the **New Supplementary Fig. 1b, d, f**.

5. Some experimental details/conclusions shall be clarified. (a) Fig1c labeled “nuclear extracts” on the top – are nuclear extracts used for ERR α only or also for other proteins? (b) Legend of Extended Data Fig 1b states that “Hepatocytes were treated with 0, 100 nM or 10 ug/ml insulin...” – please use consistent units of insulin of either nM or ug/ml in this and other figures and throughout the manuscript. (c) The authors claim that “insulin-stimulated nuclear ERR α protein was unphosphorylated (Extended Data Fig. 3a)”, but the short exposure (SE, supposedly more accurate) phospho-ERR α level seems clearly increased. (d) In Fig 3g, why do 3SA mutations affect GSK3 β S9 phosphorylation?

We thank the reviewer for raising these valid points.

(a) We used nuclear extracts for all the proteins detected for the original Fig. 1c (**Now Fig. 1e**, see below). To be consistent with other panels in Fig. 1, we re-examined the whole cell extract of the circadian sample in the revised manuscript, we also changed the light / dark phase sample time points into ZT8 and ZT 16 (4 h before and 4 h after the dark cycle, respectively), when serum insulin levels were significantly upregulated caused by increased food intake at night (See below **Fig. for Reviewer 5**).

Fig. 1e

Fig. for Reviewer 5

(b) We used insulin solution from bovine pancreas (Sigma, Cat. No. I0516), whose recommended concentration for use in cell culture is 5-10 ug/ml. Alternatively, 100 nM (0.58 ug/ml) is another common concentration used for insulin stimulation in cultured cells. Thus, we used a high (10 ug/ml) or a low (0.58 ug/ml) concentration for insulin stimulation, and found that ERR α protein is induced by insulin in a dose-dependent manner. We’ve revised the insulin units in ug/ml throughout the manuscript.

(c) We apologize for the unexplained abbreviations (SE, LE) and confusing statement. In the original Extended Data Fig.3a (see below), we could not observe an upper band shift (phosphorylated ERR α) with the nuclear extract even using longer exposure (LE), the only appeared lower band that increased by insulin should be the dephosphorylated ERR α , thus we previously stated in the main text that “insulin-stimulated nuclear ERR α protein was unphosphorylated”.

We optimized this experiment by adding whole cell extract (WCE) and cytoplasmic fraction as controls and properly labeled the bands (**New Supplementary Fig. 3a**). As shown below, in comparison with whole-cell extract, phos-tag gel examination of the

nuclear extract from HepG2 cells could not reveal an upper phosphorylated ERR α band shift, indicating that insulin-induced nuclear ERR α protein was dephosphorylated. We further explained the methods and mechanism for the phos-tag gel in the Method.

Original Extended Data Fig. 3a

Revised Supplementary Fig. 3a

(d) This is likely because ERR α transcriptionally represses the insulin signaling pathway as a feedback mechanism. When we separated the enriched pathway for ERR α -bound up- and down-regulated insulin-sensitive DEGs identified in HepG2 cells presented in original Extended Fig 1h (**Now Supplementary Fig. 1k**, see below), we observed that ERR α -bound insulin down-regulated DEGs are enriched in the PI3K-AKT signaling pathway.

Indeed, 3SA mutations affect GSK3 β S9 phosphorylation was also observed *in vivo*. As shown in **Fig. 6c** (see below), insulin-stimulated GSK3 β phosphorylation was impaired in ERR α ^{3SA} mice.

Supplementary Fig. 1k

Fig. 6c

Reviewer #3 (Remarks to the Author):

Xia and colleagues showed insulin controls the activity of ERR α via GSK3b/FBXW7 axis *in vitro*. They demonstrated ERR α is phosphorylated by GSK3b, downstream target of PI3K-AKT pathway, and ubiquitinated by FBXW7 leading to proteasome dependent

degradation. In the latter part of the manuscript, they generated mice with ERR α 3SA, and found that ERR α 3SA mice develop insulin resistance, which showed the biological significance of ERR α in vivo.

Overall, the authors have done an excellent analysis, and revealed that ERR α is one of key molecules that control energy homeostasis.

We thank the reviewer for recognizing the biological significance of our study and the constructive comments. We have carefully addressed the points raised in our revised manuscript. Please see our detailed response below.

Specific points:

1. In Extended Fig 1h, 7079 targets of ERR α ChIP-seq are shown, and it seems that these targets are from 20 kb upstream and downstream of genes. Should the targets from 20 kb downstream of genes be included in this analysis because ERR α is a transcriptional factor? Do the 7079 targets include multiple sites from one gene?

The mechanism of transcription regulation is still not fully understood. The notion of binding "upstream" and "downstream" of a transcriptional unit only makes sense when we read the DNA sequence from left to right, but folding of the genome needs to be considered, which might place regulatory elements close in three-dimensional space. There are many regulatory regions of DNA that don't follow the most common model of transcriptional activation: binding of transcriptional factor just upstream of the transcription start site (TSS) to help position RNA polymerase II at the transcriptional start site of the gene. For example, transcriptional regulation may also be achieved by distal transcription factor binding events at genomic regions termed 'enhancers'; some introns could strongly stimulate mRNA accumulation from several hundred nucleotides downstream of the TSS; transcription factor can also form transcription factor complexes with other transcription factors to affect gene transcription.

In the case of ERR α , the receptor binds to promoters but also has a strong preference for binding in introns. Overall, 48.2% of ERR α peaks (11193 of 23226) are found in introns (our previously published study PMID: 23562079, also see below **Fig. for Reviewer 6a**). Among ERR α -bound introns, there is a stronger preference for binding to the first intron (**Fig. for Reviewer 6b**). Actually, we find more peaks +10 to +20 kb compared to -10 to -20 kb of the TSS (**Fig. for Reviewer 6b**). This explains why \pm 20 kb was chosen.

There are 11777 ERR α liver ChIP-seq peaks within \pm 20 kb of a TSS, which are attributed to 7079 unique genes.

Fig. for Reviewer 6a

Fig. S3A of PMID: 23562079

Fig. for Reviewer 6b

Fig. S3B of PMID: 23562079

Fig. for Reviewer 6c

Related to PMID: 23562079

2. Alb-Cre and Gsk3b^{f/f} mice are both controls, but ERR α expressions are so different between them. This figure is confusing (Figure 2d). Also, it is better to show the data from Alb-Cre/Gsk3b^{+/f}.

For the generation of tissue-specific knockouts using the Cre/lox system, there are three recommended controls: homozygous floxed mice, Cre strain, and cre/floxed het mice (<https://www.jax.org/news-and-insights/jax-blog/2011/september/cre-lox-breeding#>).

We do agree that Alb-Cre/GSK3 β ^{+/f} littermates might be a better control that could eliminate the contribution of Cre recombinase to the phenotype of interest.

Unfortunately, we did not keep these mice, and we are unable to regenerate these mice due to time restriction during the revision. Instead, we followed the paper originally generated the GSK3 β LKO mice (PMID: 18694957) and examined the other two controls: GSK3 β ^{fl/fl} and Alb-Cre. Given both controls displayed less hepatic ERR α protein levels in comparison to the GSK3 β -null liver. These data convincingly show that GSK3 β deletion stabilizes ERR α protein.

To avoid the differences between the two controls, we have compared them with GSK3 β LKO mice separately, as shown in **New Fig. 2d** and **New Supplementary Fig. 2d**, see below.

Fig. 2d

Supplementary Fig. 2d

Correspondingly, we have revised the main text as “Indeed, ERR α protein accumulated in livers of GSK3 β liver-specific knockout (LKO) mice in comparison with their floxed littermate controls (Fig. 2d) or age-matched Alb-Cre mice (Supplementary Fig. 2d).”

3. The accumulation of ERR α protein is nicely shown in Figure 2e and 2f, however, mRNA levels should be examined to indicate that these are caused by post-translational modification, not by transcriptional level.

We agree with the reviewer that it is important to examine ERR α mRNA expression levels and have conducted the examinations for the original Fig. 2e, f (Now Fig. 2f, h), which ruled out the involvement of transcriptional regulation. Please see the **New Fig. 2f, h**, as below.

Fig. 2f

Fig. 2h

4. It is difficult to see the result in Figure 3i, upper right panel, because upper part of the picture is cut off.

We thank the reviewer for pointing this out. We revised **Fig. 3i** with the whole membrane presented for upper right panel together with molecular weight markers (see below). We previously cropped the higher ubiquitinated bands, which could potentially correspond to other ubiquitinated proteins in complex with ERR α .

Fig. 3i

5. Is there any change in endogenous ERRα protein level after MG132 treatment in HepG2 cell (Figure 3)?

We confirmed that MG132 increases endogenous ERRα protein but not mRNA in both HepG2 cells and C2C12 myoblast cells. We included these data in **New Supplementary Fig. 3d, e** and **New Supplementary Fig. 5e, f**.

Supplementary Fig. 3

Supplementary Fig. 5

6. The authors showed FBXW7 is a ubiquitin ligase of ERRα in addition to Stub1 and Parkin. Which ligase functions mainly as an ubiquitin ligase of ERRα *in vivo* (Figure 4)?

The physiologic contexts that promoted endogenous ERRα degradation by endogenous Stub1 remain unknown. Also, there is no data supporting that Stub1 regulates ERRα stability *in vivo*. While the relationship between ERRα and Parkin has been examined *in vivo*, separate studies showed different results: Parkin promotes the degradation of ERRs in the brain (PMID: 21177257) but not in the heart (PMID: 32444656), indicating tissue-specific differences. Here we first reveal that FBXW7 degrades ERRα in a phosphorylation-dependent manner *in vitro* and *in vivo*.

We also confirmed that STUB1, PARKIN, and other potential ERR α E3 ligases identified in Fig. 4 (FBXO7, FBXO11, WWP1) regulate ERR α stability independent of its phosphorylation at S19, 22, 26, as their interactions with ERR α were not affected by the phospho-defective 3SA mutations (**New Supplementary Fig. 4h**). The extent and physiological relevance of their associations with ERR α remain to be investigated.

Supplementary Fig. 4h

Which E3 mainly regulates ERR α stability *in vivo* mainly likely depends on tissue type, upstream nutrients and hormone signals. We have also modified our discussion to reflect these points.

7. There is no FBXW7 immunoblot in Figure 4c.

We thank the reviewer for pointing this out. We've included the FBXW7 immunoblot in Fig. 4c, see below.

Fig. 4c

8. In case of analysis of conditional knockout mice, controls should be Alb-Cre or Alb-Cre/Fbxw7+/f (Fig 4k).

As presented in our response to point#2 above, we agree with the reviewer that Alb-Cre/Fbxw7+/f littermate is a better control than Alb-Cre and FBXW7^{fl/fl} mice. Unfortunately, we did not keep these mice.

Instead, we followed the paper initially generated the liver-specific FBXW7 knockout mice (PMID: 21123947) and a recent paper phenotyping the FBXW7 LKO mice (PMID: 27238018) and used the Floxed littermates as the controls.

In the revised manuscript, we've further compared ERR α protein levels in the FBXW7 LKO and age-matched Alb-Cre mice and observed accumulated ERR α protein in the FBXW7-null liver comparing with the Alb-Cre mice (see below **New Supplementary Fig. 4h**). Our data convincingly show that FBXW7 negatively regulates ERR α protein stability *in vivo*.

Supplementary Fig. 4l

9. There are three isoforms in FBXW7. FBXW7 α is ubiquitously expressed, and FBXW7 γ is expressed in muscle. Which isoform is an ubiquitin ligase of ERR α in muscle (Figure 5b)?

FBXW7 α is functionally the most dominant isoform, which is ubiquitously expressed and carries out the most known FBXW7 functions (reviewed in PMID: 25314076). We observed comparable expression levels of FBXW7 α mRNA in liver and muscle (**New Supplementary Fig. 5g**). We also assessed the distribution of FBXW7 isoform mRNAs (**New Supplementary Fig. 5h**) and confirmed that FBXW7 γ is the most expressed isoform in muscle, which is consistent with previous observation (PMID: 16989775).

Supplementary Fig. 5

Despite its high expression level in muscle, FBXW7 γ remains in the nucleolus and its function is poorly understood. Also, it is unclear whether ERR α displays nucleolar localization in the muscle or whether other components of the SCF complex (Skp, Cullin, F-box containing complex) are localized in the nucleolus. Thus, FBXW7 γ may not act as an active E3 ligase for ERR α .

We have searched in commercial sources but failed to access siRNAs and plasmids targeting specific FBXW7 isoforms. Thus, it is difficult to reveal which FBXW7 isoform actively regulates ERR α stability in the muscle, although it will be interesting to define the specific effects of the two isoforms in this tissue.

10. The time points with significant difference are varied in the night time, although C29 was injected at the same time (Figure 7k). Is there any reason for this?

Various factors including fluctuations in physical activity (sleep, locomotion), food and water intake, drug absorption, and equipment noise could contribute to variations in the time points RER achieving significant difference post C29 injection. Similarly, as observed in Fig. 5k, I, RER varies at the same time points on different days. Thus, to achieve trustable measurements, we usually remain the mice in the metabolic cage for multiple days to minimize the effects caused by behavioral and metabolic alterations.

REVIEWERS' COMMENTS

Reviewer #1 (Remarks to the Author):

I think the authors have done a nice job of responding to the critiques of the initial version of their manuscript. I am pleased to recommend publication of this revised version.

Reviewer #2 (Remarks to the Author):

The authors have convincingly addressed my concerns in their revised manuscript.

Reviewer #3 (Remarks to the Author):

The authors have addressed all my questions. I recommend the manuscript for publication.

REVIEWERS' COMMENTS

Reviewer #1 (Remarks to the Author):

I think the authors have done a nice job of responding to the critiques of the initial version of their manuscript. I am pleased to recommend publication of this revised version.

Reviewer #2 (Remarks to the Author):

The authors have convincingly addressed my concerns in their revised manuscript.

Reviewer #3 (Remarks to the Author):

The authors have addressed all my questions. I recommend the manuscript for publication.

Response to NCOMMS-21-40051A comments

We are pleased that all reviewers are satisfied by our revised manuscript. We are grateful to their constructive comments and helpful suggestions, which have helped us address the issues in our manuscript and improve it.